EMBO
reports

# Transient deSUMOylation of IRF2BP proteins controls early transcription in EGFR signaling

Sina V Barysch[1,†], Nicolas Stankovic-Valentin[1,†], Tim Miedema[1], Samir Karaca[2], Judith Doppel[1], Thiziri Nait Achour[1], Aarushi Vasudeva[1], Lucie Wolf[3,4], Carsten Sticht[5], Henning Urlaub[2,6] & Frauke Melchior[1,*] (iD)

## Abstract

Molecular switches are essential modules in signaling networks and transcriptional reprogramming. Here, we describe a role for small ubiquitin-related modifier SUMO as a molecular switch in epidermal growth factor receptor (EGFR) signaling. Using quantitative mass spectrometry, we compare the endogenous SUMO proteomes of HeLa cells before and after EGF stimulation. Thereby, we identify a small group of transcriptional coregulators including IRF2BP1, IRF2BP2, and IRF2BPL as novel players in EGFR signaling. Comparison of cells expressing wild type or SUMOylation-deficient IRF2BP1 indicates that transient deSUMOylation of IRF2BP proteins is important for appropriate expression of immediate early genes including *dual specificity phosphatase 1* (DUSP1, MKP-1) and the transcription factor ATF3. We find that IRF2BP1 is a repressor, whose transient deSUMOylation on the DUSP1 promoter allows— and whose timely reSUMOylation restricts—DUSP1 transcription. Our work thus provides a paradigm how comparative SUMO proteome analyses serve to reveal novel regulators in signal transduction and transcription.

**Keywords** ATF3; DUSP1; EGFR; IRF2BP1; SUMO
**Subject Categories** Post-translational Modifications & Proteolysis; Signal Transduction

## Introduction

Small ubiquitin-related modifier (SUMO) is an essential protein modification that regulates hundreds of proteins and numerous processes including signal transduction and transcription processes (Gareau & Lima, 2010; Flotho & Melchior, 2013; Seeler & Dejean, 2017; Zhao, 2018). The last decade has seen a dramatic improvement of SUMO proteomics (as reviewed in (Hendriks & Vertegaal, 2016)), which culminated in several thousand SUMO-modified proteins and the startling number of 14,869 different SUMO2/3 acceptor sites in human cells during stress (Hendriks *et al*, 2018). Together with studies that revealed simultaneous SUMOylation of multiple subunits in protein complexes, e.g. upon DNA damage (Psakhye & Jentsch, 2012), and with the growing evidence that SUMO can contribute via low-affinity/high avidity interactions to phase separation (reviewed (Zhao, 2018)), this may lead to the impression that SUMO largely functions as a "spray" with little specificity. However, there are numerous examples where reversible SUMOylation of a single protein on a specific lysine residue determines protein function in a highly specific manner. A famous example is yeast PCNA, which interacts with the helicase Srs2 specifically upon S-phase-specific SUMOylation (Pfander *et al*, 2005). But how can we move from lists of thousands of SUMO targets to those that are most relevant? We speculated that targets whose SUMOylation changes in response to a physiological stimulus may be of particular importance.

Comparative quantitative phospho-proteomic screens have been used successfully as discovery tools to identify important players, cancer drug targets, or signaling branches, e.g. by comparing different types of cancers, wild-type versus mutant EGFR cells, or stimulation with platelet-derived growth factor PDGF versus EGF (reviewed in (Kolch & Pitt, 2010)). We thus reasoned that comparative analysis of SUMO, acting as a similar molecular switch, might be another tool to identify new key factors in signaling and signal-dependent transcription. Recently, we developed a method that allows identification and quantitative comparison of endogenously SUMOylated proteins (Becker *et al*, 2013; Barysch *et al*, 2014). This method is very well suited to identify individual proteins that may alter their SUMOylation status in response to a physiological stimulus.

For a "proof of principle" study, we turned to epidermal growth factor receptor (EGFR) signaling in HeLa cells. EGFR signaling is

1   Zentrum für Molekulare Biologie der Universität Heidelberg (ZMBH), Heidelberg University, Heidelberg, Germany
2   Bioanalytical Mass Spectrometry Group, Max Planck Institute for Biophysical Chemistry, Göttingen, Germany
3   German Cancer Research Center (DKFZ), Division of Signalling and Functional Genomics, Heidelberg, Germany
4   BioQuant & Department for Cell and Molecular Biology, Medical Faculty Mannheim, Heidelberg University, Heidelberg, Germany
5   Center of Medical Research, Bioinformatic and Statistic, Medical Faculty Mannheim, University of Heidelberg, Mannheim, Germany
6   Department of Clinical Chemistry, University Medical Center Göttingen, Göttingen, Germany
   *Corresponding author. Tel: +49 6221 546804; E-mail: f.melchior@zmbh.uni-heidelberg.de
   †These authors contributed equally to this work

one of the most prominent, best-characterized, and essential signaling networks in metazoans. It is involved in cellular growth and development, as well as in cancer progression. Key to its many roles is the transcriptional reprogramming of cells. After stimulation with EGF, a highly complex signaling network is activated, several actors work in parallel and compensatory mechanisms as well as feedback-loops are installed (Citri & Yarden, 2006; Kolch & Pitt, 2010). In the so-called early loops, phosphorylation and ubiquitylation play essential roles to control ligand-induced receptor endocytosis and cytosolic signaling events. The late loops involve transcriptional regulation, which acts in three temporal phases, the immediate early gene (IEG), the delayed early gene (DEG), and the secondary response gene (SRG) transcription waves (Avraham & Yarden, 2011).

Large quantitative studies have been performed to investigate EGF-induced changes in the phospho-proteomes (Kratchmarova *et al*, 2005; Olsen *et al*, 2006; Oyama *et al*, 2009) and the ubiquitin-proteome (Argenzio *et al*, 2011). Those studies led to the identification of important regulatory events, mainly in the early loop of the EGF response. Here we investigated changes in the endogenous SUMO1 and SUMO2/3 proteome in EGFR signaling using quantitative mass spectrometry 10 min after or without EGF treatment. As detailed below, we could indeed identify a SUMO-dependent molecular switch in EGF receptor signaling that involves the IRF2BP family of transcription coregulators. IRF2BP proteins gain increasing interest, particularly in the context of inflammation, but they have not yet been linked to EGF receptor signaling. Our findings reveal that this protein family plays an important role in immediate early gene expression and demonstrate how transient deSUMOylation of a repressor can serve to control temporal gene expression.

# Results and Discussion

### EGF induces transient deSUMOylation of several transcriptional regulators

To address the question whether EGFR signaling induces changes in the SUMO proteome and whether we are able to identify potentially new key factors in EGFR signaling, we combined our previously published method to enrich endogenously SUMOylated proteins (SUMO-IP, (Becker *et al*, 2013; Barysch *et al*, 2014)) with SILAC-based quantitative mass spectrometry. For this, labeled HeLa cells were serum-starved and treated with 0 or 100 ng/ml EGF for 10 min, respectively. This early time point was chosen to possibly detect both early events, e.g., at the plasma membrane, as well as early downstream events in transcription. Combined cell lysates were subjected to SUMO1- and SUMO2/3-IPs, followed by quantitative mass spectrometry in three independent experiments with label swapping (Fig 1A). The vast majority of identified SUMO candidates (1,228 for SUMO1 and 855 for SUMO2/3) were equally abundant in EGF-treated and untreated samples. While no protein seemed to quantitatively lose or gain SUMO after 10 min of EGF stimulation, 11 proteins could be identified whose abundance differed significantly ($P < 0.001$) between both samples (Fig 1B and C, left panel, Dataset EV1). Intriguingly, five of these were transcriptional coregulator proteins, TRIM24/TIF1α (Le Douarin *et al*, 1998) and TRIM33/TIF1γ (Venturini *et al*, 1999), as well as IRF2BP1, IRF2BP2, and

IRF2BPL (Childs & Goodbourn, 2003). To validate this finding, we repeated the SUMO-IPs and tested candidates by immunoblotting (Fig 1C, right panel). Indeed, as indicated by the decreased mobility of proteins in the IP compared to the input, IRF2BP1, IRF2BP2, and TRIM24 were SUMOylated and lost SUMO upon EGF treatment. RanGAP1 and TRIM28, two known SUMO targets whose abundance did not alter in the SUMO proteomic analysis, served as controls (Fig 1C). We next analyzed the kinetics of deSUMOylation by performing an EGF time course experiment. Surprisingly, deSUMOylation of TRIM24, IRF2BP1, and IRF2BP2 is very transient with a minimum of SUMO after 15-min EGF stimulation and full recovery after 60 min (Fig 1D).

None of our five strongest hits has previously been described to be directly involved in EGF receptor signaling, even though TRIM24 can cross-talk with PI3K/AKT signaling (Zhang *et al*, 2015; Lv *et al*, 2017) and TRIM33, as well as IRF2BP1 and IRF2BP2 are known players in TGF-β signaling (Faresse *et al*, 2008; Xi *et al*, 2011; Quéré *et al*, 2014; Manjur *et al*, 2019; Yuki *et al*, 2019). Furthermore, while TRIM24 and TRIM33 have been shown to be SUMOylated (Seeler *et al*, 2001; Fattet *et al*, 2013), SUMOylation of IRF2BP1, IRF2BP2, and IRF2BPL had not been investigated yet (see below).

### IRF2BP proteins are SUMOylated at their conserved C-termini

IRF2BP proteins are transcriptional coregulators that can homo- and hetero-oligomerize via a conserved N-terminal zinc finger and interact with diverse transcription factors via a C-terminal RING domain (Childs & Goodbourn, 2003; Yeung *et al*, 2011). They have initially been identified in a yeast two hybrid screen as interaction partners of IRF2 (Childs & Goodbourn, 2003). In the following years, several groups described individual target genes that they transcriptionally regulate, such as the TGF-β-Smad target genes ADAM12 and p21Cip1 (Faresse *et al*, 2008), as well as VEGFA (Teng *et al*, 2010) and FASTKD2 (Yeung *et al*, 2011). More recently, a ChIP-Seq study of IRF2BP2 in mouse MEL cells revealed more than 11,000 binding sites in the mouse genome (Stadhouders *et al*, 2015).

Very little is known about IRF2BP1's biological functions, but IRF2BP2 has emerged as an important factor in the immune system. It can suppress inflammation in macrophages and microglia (Chen *et al*, 2015; Cruz *et al*, 2017; Hari *et al*, 2017) and has inhibitory effects on the expression of the programmed death ligand 1 (PD-L1) (Soliman *et al*, 2014; Dorand *et al*, 2016; Wu *et al*, 2019). In addition, IRF2BP2 was recently found to regulate the Hippo pathway and to act as a tumor suppressor in hepatocellular carcinoma (Feng *et al*, 2020).

To investigate functional consequences of EGF-dependent deSUMOylation of IRF2BP proteins, we sought to identify their SUMO acceptor sites. SUMO modifies its target proteins with the help of SUMO-specific E1, E2, and E3 enzymes on lysine residues that are typically embedded in SUMO consensus sites (ΨKxE/D, where Ψ is a bulky hydrophobic residue), as reviewed in (Gareau & Lima, 2010; Flotho & Melchior, 2013). When we inspected IRF2BP1 and IRF2BP2 for putative SUMOylation sites, we found two motifs that were conserved within the family and between species, a minimal KxE motif (FKKD/E) in the otherwise poorly conserved middle region and a classic SUMO consensus site (VKKE) close to the C-terminus (Fig 2A). To test whether one of those predicted IRF2BP SUMOylation sites is indeed the functional

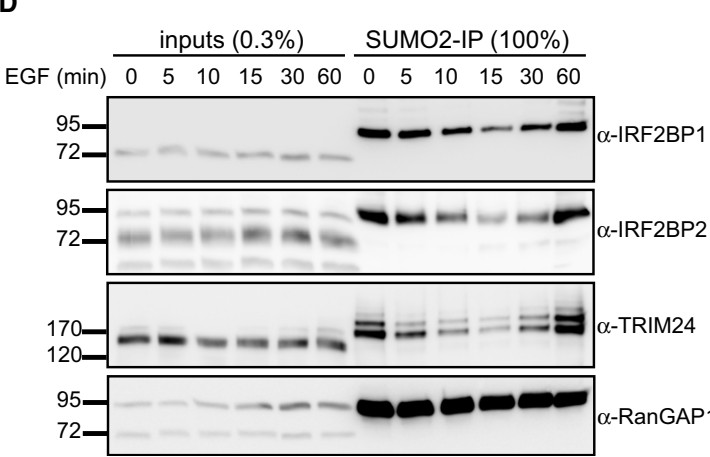

**Figure 1.**

**Figure 1. EGF induces rapid and transient deSUMOylation of transcriptional regulators in HeLa cells.**

A   Schematic representation of a quantitative proteome analysis that compares the SUMO proteome of serum-starved HeLa cells treated for 10 min with or without 100 nM EGF (SILAC labeling and endogenous SUMO1- and SUMO2/3-IPs).
B   Scatterplots represent SILAC/SUMO-IP quantification after EGF treatment from three biological replicates. Each dot represents a protein that is either present in the same ratio between untreated and EGF-treated cells (dark blue dots) or is significantly more present in one of the samples (red and yellow dots).
C   Left panel: Bar graph depicting mass spectrometry results of 11 proteins with altered SUMOylation (significant hits with $P < 0.0001$), as well as the three non-changing proteins SUMO2, RanGAP1, and TRIM28. Right panel: Proteins highlighted in bold in the left panel were validated by SUMO-IP/immunoblotting from serum-starved HeLa cells without or with 10-min EGF stimulation.
D   Time course experiment: Serum-starved HeLa cells were treated with 100 nM EGF, and samples were harvested at indicated times and subjected to SUMO2 IP followed by immunoblotting with the indicated antibodies. IRF2BP1, IRF2BP2, and TRIM24 are rapidly but transiently deSUMOylated upon EGF treatment.

one, we transiently transfected HA-tagged wild-type and KR mutant variants of both proteins into HeLa cells and performed endogenous SUMO-IPs. While SUMOylated IRF2BP wild type and K247R or K326R mutants can be immunoprecipitated, no signal was detected for the C-terminal K579 or K566R mutants. Thus, the lysine within the predicted C-terminal consensus SUMO site seems to be the dominant one endogenously used in mammalian cells (Fig 2B). This SUMOylation site is located at the very C-terminus of IRF2BP proteins, within a stretch of 30 highly conserved amino acids that follow directly after the conserved RING domain (Fig 2C). Of note, this SUMOylation site was not identified in recent proteomic screens from human cells (Hendriks *et al*, 2014; Impens *et al*, 2014; Xiao *et al*, 2015; Hendriks *et al*, 2018), likely because the branched peptides that were generated upon protease digest of IRF2BP proteins were too short to be assigned. However, very recently, IRF2BP2 SUMOylation on this conserved lysine residue was found in zebrafish (Wang *et al*, 2020). Taken together, in HeLa cells, IRF2BP proteins are SUMOylated at a very conserved lysine residue, which is located at their C-terminus.

## SUMOylation-deficient IRF2BP1 cells differ in EGF-dependent transcription

To gain insights into the functional consequences of IRF2BP protein (de)SUMOylation and its role in EGFR signaling, we next generated stable cell lines expressing either IRF2BP1 wild type or the SUMOylation-deficient mutant. To avoid problems arising from variable expression levels and from tags that were reported to interfere with IRF2BP function (Giraud *et al*, 2014), we generated stable polyclonal HeLa cell lines that express untagged and siRNA-resistant variants of IRF2BP1 (wild type or K579R) using a bi-cistronic vector system (pIRES-hrGFPII; Fig 2D). GFP selection by FACS was used to specifically select low expressing cells, thus leading to exogenous IRF2BP1 expression that matches endogenous protein levels (Fig 2E). Expression levels of exogenous wild type and K579R IRF2BP1 were comparable, suggesting that they have similar stability. Furthermore, localization of exogenous IRF2BP1 was similar in both cell populations (Fig 2F), indicating that SUMO does not regulate nucleocytoplasmic transport of these proteins. Another function that has been attributed to SUMO is an influence on transcription factor—chromatin interaction (reviewed, e.g., in (Rosonina *et al*, 2017)). We therefore compared chromatin binding of wild-type IRF2BP1 with its SUMO-deficient mutant in salt extraction experiments. A significant fraction of IRF2BP1 binds stably to chromatin, and no difference could be observed between wild-type and mutant forms and also not for SUMOylated wild-type IRF2BP1, which is visible in extracts of stable cell lines (Fig 2G).

To begin to address the question whether SUMOylation of IRF2BP1 may contribute to IRF2BP's role in gene expression, we next used microarrays to compare the transcriptome between wild-type and mutant cells that were depleted of endogenous IRF2BP1 and grown asynchronously for 48 h in serum containing medium (Fig EV1A). Indeed, 138 genes were at least 1.5 fold differentially expressed between the two cell lines (see lists of genes in Dataset EV2). Gene Ontology analysis revealed that many of these genes are involved in the regulation of cell adhesion, proliferation, and in the response to growth factor stimuli (Fig EV1B).

The most important question was, however, whether the lack of IRF2BP1 SUMOylation—and in consequence the lack of temporally controlled deSUMOylation—would result in transcriptional changes upon EGF stimulation. To address this, we again depleted endogenous IRF2BP1 from our polyclonal wild-type IRF2BP1 and K579R cell lines, serum-starved the cells for 16 h, and incubated them for an additional hour with or without 100 ng/ml EGF, before cells were harvested and their RNAs quantified using microarray analyses (Fig EV1A, Dataset EV2). As in full serum, genes that were differentially expressed between wt and mutant cells clustered in GO categories such as cell adhesion, proliferation, and in the response to growth factor stimuli (Fig EV1C and D for the absence and presence of EGF, respectively). Consistent with studies in HeLa cells from Yarden and coworkers (Amit *et al*, 2007), we identified 529 genes significantly regulated by EGF in our stable cell lines (at least 1.5-fold in IRF2BP1 wild-type or in IRF2BP1 KR cells). Intriguingly, for 38 (7%) of those EGF-responsive genes, a significant difference in the amplitude of the transcriptional change induced by EGF could be observed between wild-type and KR cells (at least 1.5 fold, Fig 3A). In light of the early time point (1 h after EGF treatment), we considered it likely that these 38 genes are directly regulated by IRF2BP1.

## The feedback regulator DUSP1 is a direct target of IRF2BP1

IRF2BP proteins are transcriptional coregulators that seem to interact with many different transcription factors and coregulators. In mouse MEL cells, 40% of the IRF2BP2 binding sites were found within 5 kb of the transcriptional start sites (Stadhouders *et al*, 2015) and IRF2BP2 bound > 2,000 genes in their proximal promoter region (Fang *et al*, 2020). We thus asked whether any of our 38 candidate genes are among those genes. Indeed, seven of the 38 genes, including DUSP1 (MKP-1), activating transcription factor 3 (ATF3), Fos and early growth response protein 2 (Egr2), interacted in mouse MEL cells with IRF2BP2 in the proximal promoter region.

DUSP1 is a well-known immediate early gene and its gene product, dual specificity phosphatase 1, is an inhibitor of the MAP kinase branch of EGF receptor signaling and thus an important

feedback regulator in EGFR signaling (reviewed in (Liu *et al*, 2007)). In consequence, DUSP1 seemed an excellent candidate to gain mechanistic insights into how IRF2BP (de)SUMOylation contributes to gene expression. We thus tested by chromatin IP experiments whether IRF2BP2 and IRF2BP1 bind to the human DUSP1 promoter in HeLa cells. Indeed, both proteins bind to a region (-243 to -67) directly adjacent to the TSS (Fig 3B and C). To validate a second

candidate, we also confirmed binding of IRF2BP1 and IRF2BP2 to the proximal promoter of human ATF3 in HeLa cells (Fig EV2). ATF3 was a particularly promising candidate, because it had just been found to be under direct control of IRF2BP2 in mouse non-alcoholic fatty liver (Fang *et al*, 2020).

We next wanted to interrogate whether transient deSUMOylation of IRF2BP proteins regulates its association with the DUSP1

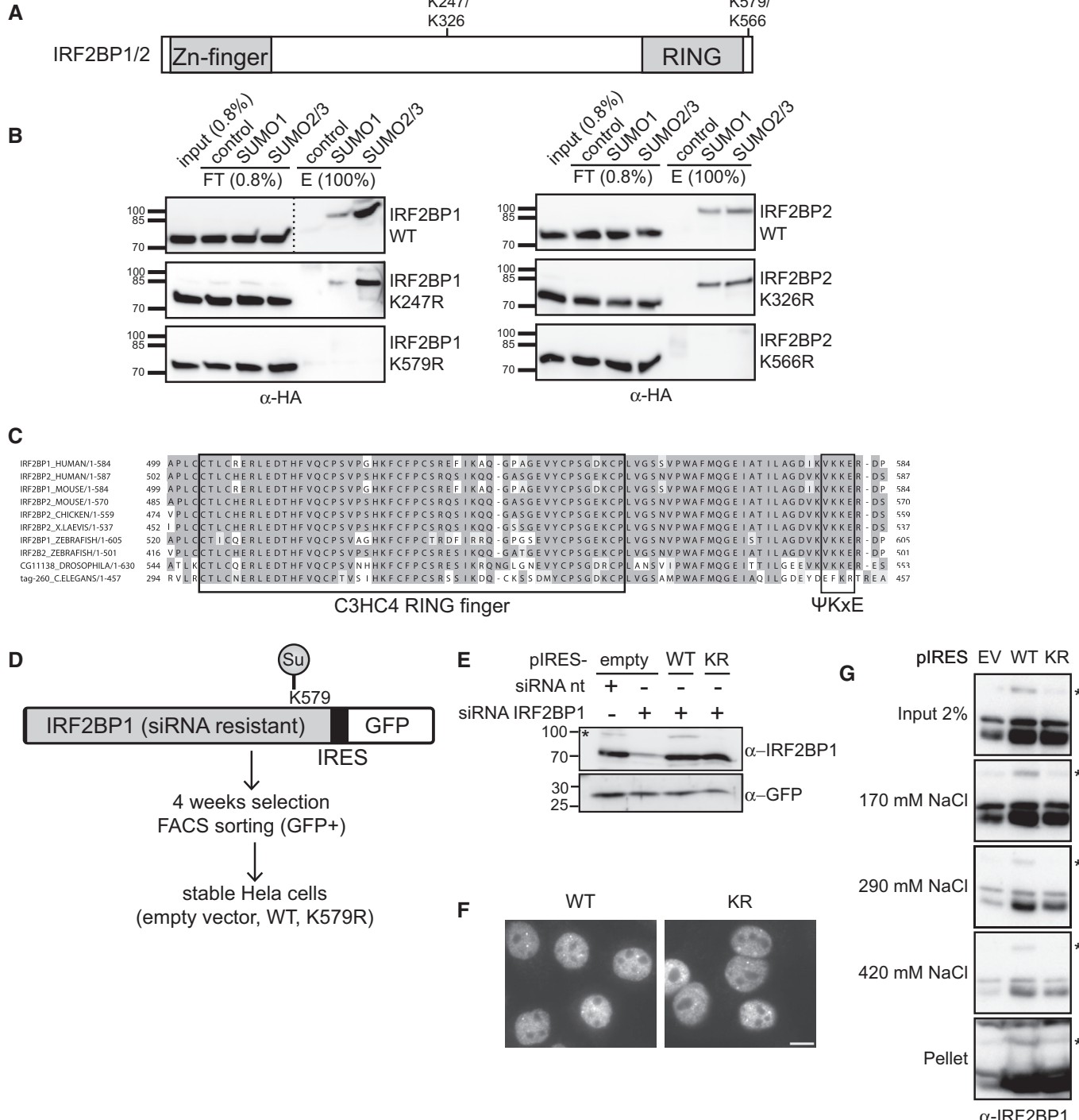

**Figure 2.**

◀

**Figure 2.  IRF2BP proteins are SUMOylated at their highly conserved C-terminus.**

A   Schematic representation of the domain structure of IRF2BP1 and IRF2BP2. The primary sequence suggests two putative SUMO sites that are conserved among different species.
B   Identification of the endogenous SUMO sites in IRF2BP1 (left panel) and IRF2BP2 (right panel). HeLa cells were transfected with HA-tagged wild-type (WT) or mutant (KR) proteins, endogenous SUMO-IPs were performed, and the HA-signal was analyzed by immunoblotting. Mutation of the C-terminal SUMO site abolished SUMOylation of IRF2BP1 and IRF2BP2 in HeLa cells.
C   Clustal omega analysis of IRF2BP1 and IRF2BP2 from *Homo sapiens, Mus musculus, Gallus gallus, Xenopus laevis, Danio rerio, Drosophila melanogaster* and *Caenorhabditis elegans* shows high conservation of the C-terminal region including the SUMO site.
D   Schematic representation of the creation of stable, untagged, and siRNA-resistant IRF2BP1 WT and K579R HeLa cells. Constructs expressing IRF2BP1 variants in an pIRES-hrGFP II ("pIRES") vector were transfected, selected with antibiotics, and FACS sorted for low GFP expression.
E   Stable HeLa cells expressing pIRES-empty vector, IRF2BP1 WT, or IRF2BP1 K579R were treated with siRNA against endogenous IRF2BP1 or non-targeting (nt) siRNA. Exogenous siRNA-resistant IRF2BP1 was expressed at low levels similar to endogenous IRF2BP1. * refers to an unspecific band.
F   Wt and mutant IRF2BP1 localizes in the nucleus. After knockdown of endogenous IRF2BP1, stable IRF2BP1 (WT or K579R) cell lines were immunostained for IRF2BP1. Exogenous IRF2BP1 variants show a similar nuclear localization. Scale bar = 10 μm.
G   IRF2BP1 wt and mutant associate with chromatin to a similar extent. HeLa cells were lysed in 0.075% NP40 (Input). After centrifugation, the nuclei were incubated and vortexed with a nuclear extract (NE) buffer containing 170 mM NaCl. The eluates were collected, and the procedure was repeated using a NE buffer with higher salt concentrations, first 290 mM, then 420 mM. Wild-type IRF2BP1, the SUMO-deficient K579R mutant and the SUMOylated wild-type form (*) all behave similarly.

promoter. This does not seem to be the case: Association with the DUSP1 promoter does neither differ significantly between IRF2BP1 wild type and the K579R mutant, nor is it affected by EGF treatment, which causes IRF2BP1 deSUMOylation (Fig 3D). In conclusion, IRF2BP1 binding directly upstream (within 240 nucleotides) of the DUSP1 TSS is independent of its SUMOylation status and is not altered by EGF.

## IRF2BP1 is a SUMO-dependent repressor of immediate early genes

Does transient deSUMOylation of IRF2BP1 contribute to timing or amplitude of DUSP1 mRNA expression after EGF treatment? To address this question, we turned to qPCR experiments (Fig 3E). Indeed, although DUSP1 mRNA levels showed the expected time course of induction and decline after EGF treatment in wild-type IRF2BP1 and in IRF2BP1 K579R cells, the amplitude of DUSP1 expression was significantly enhanced in the mutant cell line (Fig 3E). To ensure that the observed effect was due to lack of SUMOylation, rather than other lysine modifications, we repeated these experiments with cells in which we replaced endogenous IRF2BP1 with IRF2BP1 V578A. V578A is part of the SUMOylation consensus motif and required for IRF2BP1 SUMOylation (Fig 3F, right panel). Also in these cells, the amplitude of DUSP1 transcription after stimulation with EGF was increased compared to cells expressing wild-type IRF2BP1 (Fig 3F, left panel), strongly supporting the notion that SUMOylated IRF2BP1 inversely correlates with DUSP1 expression. As pointed out above, DUSP1 is just one of several immediate early genes (IEGs) that seem to be under EGF and IRF2BP1 control. We thus repeated the qPCR analyses for three additional genes (Fig 3G). Indeed, ATF3, Egr2, and Fos were induced more rapidly and to higher amplitude in cells that expressed the SUMOylation-deficient V578A variant of IRF2BP1. In conclusion, our analyses identified IRF2BP1 as a novel SUMO-dependent regulator of several immediate early genes.

How does transient IRF2BP1 deSUMOylation contribute to timely expression of several immediate early genes? We envisioned two different explanations: Either IRF2BP1 is a repressor that requires SUMOylation for its repressive function. Or IRF2BP1 is required for transcription, but is inactive as long as it is SUMOylated. To distinguish between these scenarios, we asked whether IRF2BP1

knockdown stimulates or prevents DUSP1 and ATF3 induction. As shown by immunoblotting in Fig 4A, knockdown of IRF2BP1 prior to serum starvation and EGF treatment caused a strong increase in DUSP1 and in ATF3 induction, consistent with a repressive function.

Many different mechanisms have been described for SUMO-dependent repression in the literature (Gill, 2005; Hay, 2005; Guo *et al*, 2007; Lyst & Stancheva, 2007; Rosonina *et al*, 2017; Rosonina, 2019). We consider it most likely that SUMO regulates IRF2BP1 interactions in the context of specific promoters. Its binding partners may for example be HDAC containing corepressor complexes, which have frequently been linked to SUMO-dependent repression (reviewed in Rosenfeld, 2006; Ouyang & Gill, 2009). In line with this idea is a publication that describes extensive IRF2BP2/NCoR co-occupancy in mouse MEL cells and activation of IRF2BP2-repressed genes upon HDAC2/3 inhibition (Stadhouders *et al*, 2015). Alternatively, SUMO may regulate interactions of IRF2BP1 with components of the transcription machinery that regulate transcription initiation or pausing: The DUSP1 gene shows strong enrichment of polymerase II at the TSS in serum-starved HeLa cells (Gardini *et al*, 2014), and its expression is rapidly induced in response to EGF. How its transcription is induced remains unclear, but it could involve relieve from transcription pausing, as has been suggested by (Ryser *et al*, 2004), or an increase of transcription initiation (Ehrensberger *et al*, 2013). The DUSP1 promoter binds numerous transcription factors (ENCODE database, e.g. (Johansson-Haque *et al*, 2008)), and the proximal DUSP1 promoter of resting rat pituitary cells is enriched with transcription elongation factors such as NELF and DSIF (Ryser *et al*, 2004; Fujita *et al*, 2009). Moreover, SUMO has recently been suggested to contribute to transcriptional pausing, at least in the context of severe heat stress (Niskanen & Palvimo, 2017). Considering that SUMOylated IRF2BP1 sits directly adjacent to the TSS, it is conceivable that SUMO may contribute to the stability of a paused state or inhibit transcription initiation.

Our findings identify IRF2BP1 as a SUMO-dependent repressor of DUSP1 and other immediate early genes in EGF receptor signaling (Figs 4B and EV3). It is, however, important to note that IRF2BP1 deSUMOylation is not sufficient to drive DUSP1 expression: We do not observe premature expression or expression in the absence of EGF in the IRF2BP1 SUMO-deficient cells. At least one additional EGF-dependent event is required, which may for example be MAP

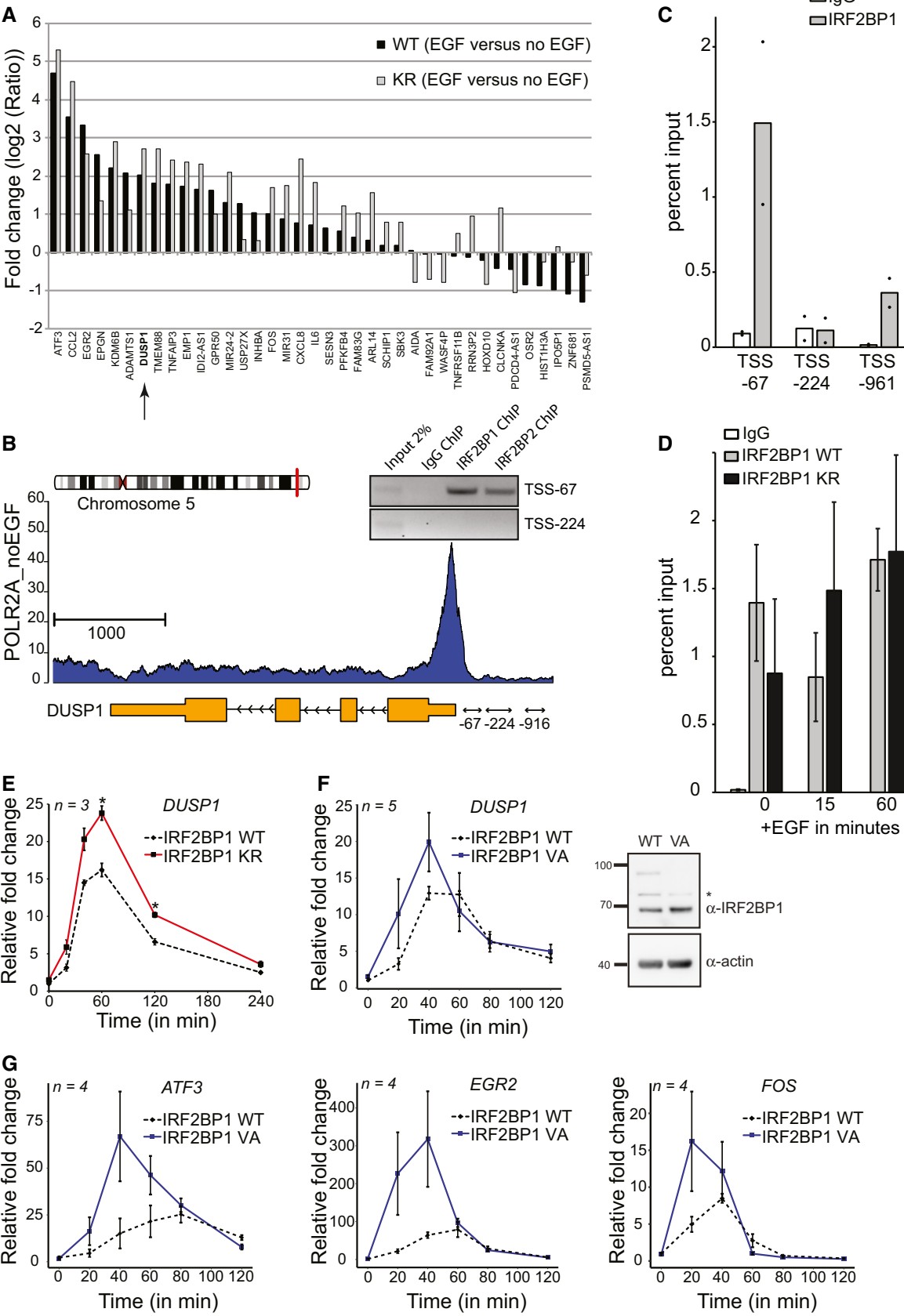

**Figure 3.**

◀

**Figure 3. IRF2BP1 deSUMOylation contributes to transcriptional induction of immediate early genes.**

A   Stable IRF2BP1 cell lines (knocked down for endogenous IRF2BP1) were used to perform a microarray experiment under serum starvation and upon treatment with EGF for 1 h. A subset of 38 EGF-dependent genes is differentially regulated in IRF2BP1 wild-type cells compared to IRF2BP1 K579R cell lines (at least 1.5-fold). Among them are DUSP1 (arrow), ATF3, Fos, and Egr2. Each microarray was performed in triplicates, and the bars show $\log_2$ values of the fold changes for wt cells (black bars) and for KR cells (gray bars). For details, see Materials and Methods.

B   Chromatin IP reveals association of human IRF2BP1 with the proximal DUSP1 promoter in HeLa cells. Gene architecture of human DUSP1. The primers at -243/-67 ("-67"), -473/-224 ("-224"), and -1,170/-961 ("-916") relative to the TSS were used for ChIP experiments. IRF2BP1 and IRF2BP2 bind to the promoter region of DUSP1 between nucleotides -243 and -67.

C   ChIP/qPCR experiments reveal preferential IRF2BP1 binding to the promoter region of DUSP1 between -243 and -67. Data show mean (bar) and individual data points from two biological replicates.

D   IRF2BP1 wild type and the K579R mutant both bind the DUSP1 promoter in the absence or presence of EGF. Stable cell lines expressing IRF2BP1 wild type or K579R were knocked down for endogenous IRF2BP1, followed by IRF2BP1 ChIP and DUSP1 qPCR of its promoter region -243 and -67. Data show means ± SEM from three biological replicates.

E   qPCR data after EGF treatment in IR2BP1 wild type and K579R cell lines after knockdown of the endogenous IRF2BP1. Data show means ± SEM from three biological replicates.

F   IRF2BP1 wild type and V578A cell lines after knockdown of the endogenous IRF2BP1 were analyzed for DUSP1 transcription after EGF treatment by qPCR (left panel), data show means ± SEM from five biological replicates (left panel), and IRF2BP1 protein levels. As expected, IRF2BP1 V578A is not SUMOylated (right panel). * refers to an unspecific band.

G   qPCR data for immediate early genes after EGF treatment in the IR2BP1 wild type and V578A cell lines after knockdown of the endogenous IRF2BP1. Data show means ± SEM from four biological replicates.

kinase-dependent phosphorylation of a transcription—or elongation regulator. SUMO thus seems to serve as an important brake that can prevent both, erroneous activation, and overshooting response.

But how can SUMO restrict DUSP1 expression, if most IRF2BP1 is not SUMOylated in cells? We envision two scenarios: Either IRF2BP1 is quantitatively SUMOylated on the DUSP1 promoter, but not on many other genes to which it binds. Alternatively, the whole pool of IRF2BP1 undergoes constant cycles of SUMOylation and deSUMOylation, and EGF shifts the equilibrium to the unmodified form. Irrespective of whether the SUMOylated species contributes to the stability of a paused state or blocks transcription initiation, shortening the lifetime of the SUMOylated species would increase the amplitude of transcription. Either model depends on a signaling event that will inhibit SUMOylation—or stimulate deSUMOylation of IRF2BP1. Very few SUMOylated proteins lose SUMO in response to EGF—we consider it therefore likely that IRF2BP1 itself, rather than one of the SUMO enzymes, is altered.

Precedence for this comes from studies of the transcription factor Elk1, whose repressive activity depends on SUMO. PMA-induced activation of MAP kinases leads to Elk1 phosphorylation and deSUMOylation, which turns Elk1 into an activator (Yang *et al*, 2003). IRF2BP1 carries numerous phosphorylation sites, none of which is close to the SUMOylation consensus site. Whether (de) phosphorylation of any of these is responsible for the EGF-dependent SUMO switch awaits further investigation. Importantly, our study supports the notion that transient deSUMOylation of transcription factors could be a commonly used mechanism for transcriptional control.

## IRF2BP proteins act as negative regulators of EGFR signaling

Although we found clear transcriptional changes in cells that express SUMOylation-deficient IRF2BP1, DUSP1 and ATF3 protein induction did not vary enough between wt and mutant cell lines to be statistically significant and reproducible. Variability may be due in part to technical reasons, for example because knockdown efficiency of the endogenous IRF2BP1 influences DUSP1 induction. More important is the possibility that loss of a single SUMO site in

IRF2BP1 may be compensated by the SUMOylation of its family members and binding partners IRF2BP2 and IRF2BPL. To gain full insights into the physiological consequences of the transient deSUMOylation of IRF2BP proteins, we would either need to interfere with an unknown upstream signal or we would have to generate cell lines in which all three proteins would be SUMOylation-deficient. Unfortunately, this is currently not possible—even mild IRF2BP1 overexpression is not well tolerated, and our stable cells lose expression of full-length IRF2BP1 within a few weeks.

We thus turned our attention to the more straight forward question of cellular consequences of IRF2BP1 loss. As discussed above, its depletion prior to EGF stimulation leads to robust DUSP1 and ATF3 upregulation (Fig 4A), two important players with rather different functions in EGFR signaling. We thus wondered how knockdown might influence cell migration and proliferation, well-known outcomes of EGF receptor signaling. As shown in Fig 4C and D, knockdown of IRF2BP1 caused both, faster EGF-dependent wound closure and accelerated cell proliferation. Knockdown of IRF2BP2 also enhanced cell proliferation, albeit not as strongly as knockdown of IRF2BP1. These findings suggest that IRF2BP proteins, especially IRF2BP1, have growth inhibitory functions.

To gain insights into possible reasons, we performed microarray analyses after knockdown of IRF2BP1, IRF2BP2, or both proteins in asynchronously growing HeLa cells (Figs 4E and EV4A). In light of the large number of genes that interact with IRF2BP2, we were not surprised to find that many genes were affected by knocking down IRF2BP1, IRF2BP2, or both: In total, approximately 5,200 genes show a significant change (> 1.5-fold, FDR < 0.05, Dataset EV3). Intriguingly, IRF2BP proteins seem to only partially control the same genes and where they do, they may even have opposing effects (Figs 4E and EV4B). The most striking finding was, however, that genes affected by IRF2BP1 and IRF2BP2 knockdown enriched strongly in diverse signal transduction categories including EGF receptor signaling ("ERBB_signaling", Figs 4F and EV4C and D for IRF2BP1 and IRF2BP2 knockdown, respectively). Two genes were particularly intriguing: The EGF receptor and Sec24D, a COPII component that facilitates transport of newly synthesized EGFR to the plasma membrane (Scharaw *et al*, 2016), were both upregulated

   

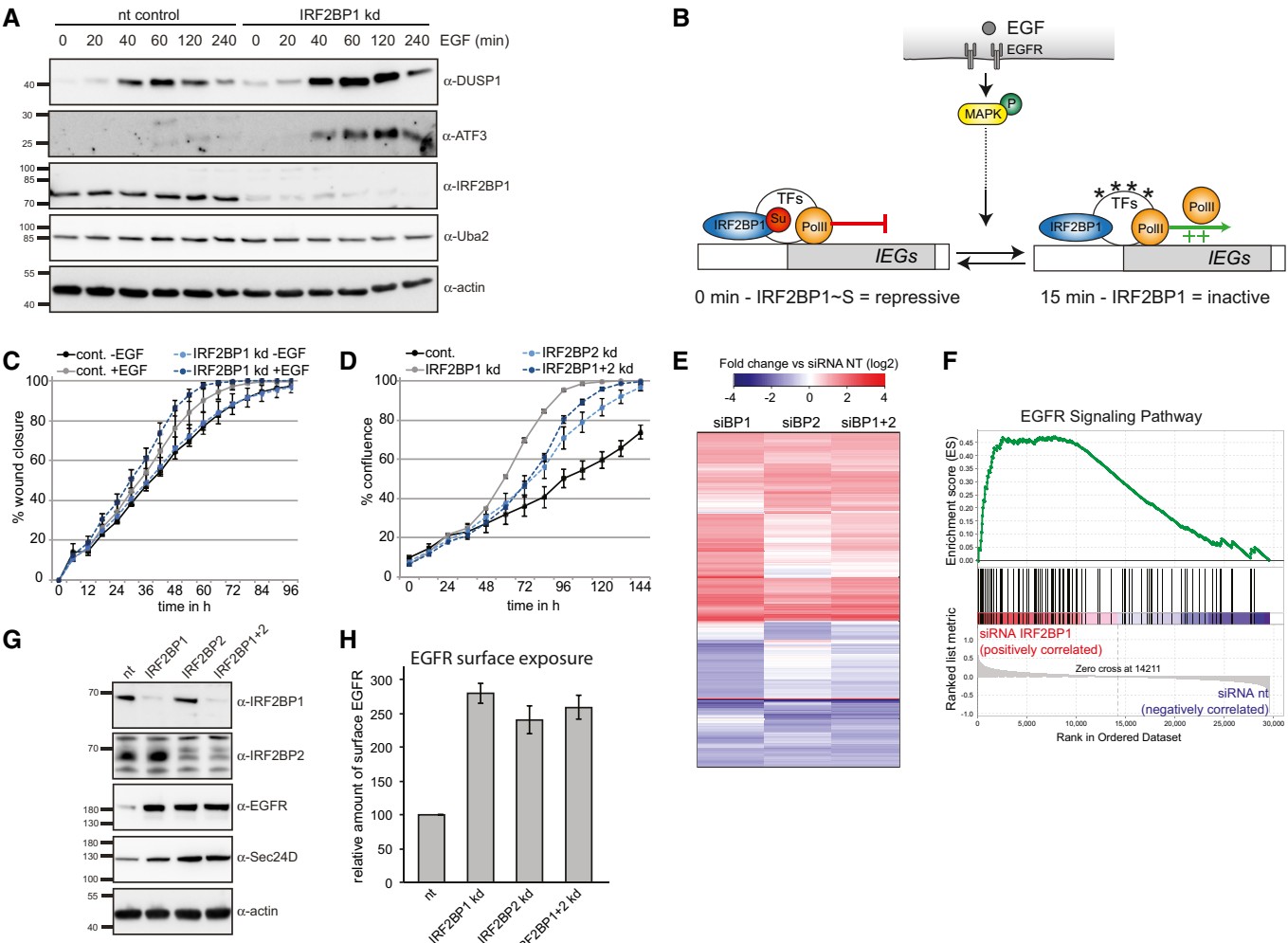

**Figure 4. IRF2BP proteins negatively regulate expression of the EGF receptor.**

A  Immunoblotting of HeLa lysates at indicated times after EGF treatment reveals enhanced DUSP1 and ATF3 expression upon knockdown of IRF2BP1. Uba2 and beta-actin serve as independent loading controls.

B  Model. IRF2BP1 is a SUMO-dependent transcriptional repressor of immediate early genes (IEGs). Transcription of IEGs is repressed by SUMOylated IRF2BP1, which binds to its proximal promoter. EGF receptor signaling yields at least two signals to induce IEG expression, one of which is the transient deSUMOylation of IRF2BP1.

C  Wound healing assay: HeLa cells were incubated with nt or IRF2BP1 siRNA for 2 days, grown to 90% confluency, and analyzed for wound closure with or without addition of 100 ng/ml EGF. Data show means of the relative wound density ± SEM from four biological replicates.

D  Proliferation assay: HeLa cells upon knockdown of IRF2BP1 and IRF2BP1 were analyzed for cell density over 6 days in growth medium. One representative biological experiment is shown with means ± SEM from five technical replicates.

E  IRF2BP1, IRF2BP2, and IRF2BP1 + IRF2BP2 were knocked down in HeLa cells for 72 h, and gene expression data were recorded by microarray analysis; non-targeting siRNAs were used as a control. IRF2BP1 and IRF2BP2 have distinct and overlapping functions. Experiments were done in triplicates.

F  GSEA of the highly significant enriched EGFR signaling pathway. IRF2BP1 knockdown correlates with an increase of genes involved in EGFR signaling.

G  Validation of two candidate genes: IRF2BP1 and IRF2BP2 were knocked down for 72 h in HeLa cells, and cell lysates were analyzed by immunoblotting with the indicated antibodies.

H  FACS-based analysis of EGFR surface expression: IRF2BP1 and IRF2BP2 were knocked down for 48 h in HeLa cells, and cells were serum-starved for 16 h, stained with fluorescent anti-EGFR antibodies, and analyzed by flow cytometry. Shown are means of three independent experiments ± SEM.

4-fold on RNA level upon IRF2BP1 and IRF2BP2 knockdown. Consistent with this, immunoblotting revealed that IRF2BP1 and/or IRF2BP2 knockdown leads to very clear upregulation of EGFR and Sec24D proteins (Fig 4G). As shown by fluorescence activated cell sorting (FACS), this also leads to strongly elevated levels of the EGF receptor at the plasma membrane (Fig 4H), which may well contribute to enhanced proliferation of IRF2BP1 and IRF2BP2 knockdown

cells in response to EGF. Upregulation of EGF receptor transcription is typically a late response, serving to replenish receptors, which have been degraded upon signaling. At presence, we have no evidence that IRF2BP proteins bind the EGF receptor gene directly. Its expression could be affected by any of the immediate early and many subsequent feed-forward and feedback regulatory events in EGF receptor signaling that may be altered upon IRF2BP

knockdown. Irrespective of the underlying mechanism, our findings reveal a thus far unknown role for IRF2BP1 in the control of growth factor signaling and cell proliferation. Consistent with this, IRF2BP1 expression has recently been suggested as a favorable prognostic marker for renal and pancreatic cancer patients (www.proteinatlas.org, (Uhlén *et al*, 2017)) and IRF2BP2 was reported to be a favorable prognostic marker for hepatocellular carcinoma (Feng *et al*, 2020).

In conclusion, our work provides a paradigm for endogenous SUMO proteomics as a discovery tool for novel players in signal transduction and it puts the spotlight on the poorly studied transcriptional regulatory protein IRF2BP1 as a novel regulator of EGF receptor signaling. Although we have not been able to fully address this, we assume that IRF2BP1 and its binding partners IRF2BP2 and IRF2BPL cooperate on immediate early promoters and that their simultaneous deSUMOylation is required for maximal effects. With the "SUMO switch" discovered here, we also add an important element that helps understanding temporal control in immediate early gene expression.

# Materials and Methods

Detailed information on all used resources and reagents (including catalogue numbers) is provided in Table EV1.

## Cell culture

### Culture of HeLa cells

HeLa cells were obtained from Dr. Simona Polo (Milan) and were the same ones as used in the studies on phospho- and ubiquitin-proteomes after EGF treatment (Olsen *et al*, 2006; Argenzio *et al*, 2011). Cells were grown in Dulbecco's modified Eagle's medium (DMEM), supplemented with 10% fetal bovine serum (FBS), 2 mM L-glutamine, and 1 mM sodium pyruvate medium (we also refer to it as "full medium"). Cells were split between 1:3 and 1:15 and were grown at 37°C, 5% $CO_2$, and 90% humidity.

### Culture of hybridoma cells

Hybridoma cells producing antibodies against SUMO1 (clone 21C7) and SUMO2/3 (Clone 8A2), both obtained from the Developmental Studies Hybridoma Bank and originally developed by Dr. Mike Matunis. Cells were cultured in CELLine 1000 bioreactor (Integra) according to manufacturer's instructions. The medium compartment was filled with RPMI medium containing 10% FBS (heat-inactivated at 56°C for 30 min) and 2 mM L-glutamine. For the cell compartment, Hybridoma-SFM (Gibco) medium was used.

### siRNA transfection of HeLa cells

HeLa cells were reverse-transfected for 3 days using OptiMEM and Lipofectamine RNAiMAX (Invitrogen) according to the manufacturer's instructions. Oligos targeting IRF2BP1 (GCUUCAAGUACC UCGAAUA[dT][dT] and UAUUCGAGGUACUUGAAGC[dT][dT]), IRF2BP2 (GAGAGAGACUCGUGACUUU[dT][dT] and AAAGUCACG AGUCUCUCUC[dT][dT]) as well as "non-targeting" (AAUAGCGACU AAACACAUCAA[dT][dT] and UUGAUGUGUUUAGUCGCUAUU[dT] [dT]) were ordered at Ambion or Applied Biosystems (both Thermo Fischer Scientific). siRNA stocks were kept at 20 µM in water at −20°C for up to 1 year.

### Plasmid transfection of HeLa cells

HeLa cells were transiently transfected 24 h after seeding at a confluency of 50% using polyethyleneimine (PEI, PolySciences (23966), 1 mg/ml pH 7.0) at a DNA:PEI ratio of 1:2.5 in serum-free DMEM. Fifteen-cm plates were transfected with 30 µg DNA and 10 cm plates with 13.5 µg plasmid DNA. Serum supplemented DMEM was added 6 h after transfection. Cells were harvested 24–48 h after transfection.

### Generation of polyclonal stable cell lines

Plasmids "pIRES-hrGFPII", "IRF2BP1 wild type, siRNA resistant, in pIRES-hrGFPII", "IRF2BP1 K579R, siRNA resistant, in pIRES-hrGFPII" or "IRF2BP1 V578A, siRNA resistant, in pIRES-hrGFPII" were transfected on 10-cm plates. After 3 days, cells were split 1:3 and full medium (supplemented with G418) was added. We routinely checked G418 efficiency and used the concentration at which all untransfected HeLa cells died within 5–7 days (varied between 0.5–1.5 mg/ml depending on the age of stock solutions).

After 2 weeks, "low GFP"-positive cells were sorted using a BD FACSAria Illu™. Usually, only 0.5–5% of the initial cell pools were "low GFP" expressing. FACS sorting was thus repeated several times until the cells appeared stable. We analyzed our polyclonal cell lines once a month by FACS and used them for experiments only if they were at least 70% low GFP expressing.

### Preparation of cells for FACS sorting

HeLa cells growing on a 10 cm dish were detached very well using 500 µl trypsin for 5 min. 500 µl quenching solution (PBS, 1% dialyzed FBS) was added, and cells were resuspended thoroughly. This cell suspension was mixed with 3 ml dissociation buffer (5 mM EDTA, 25 mM HEPES, 1% dialyzed FBS in PBS, pH 7.4), and 5 mM $MgCl_2$ and DNase (25 U/ml cells) were added. Cells were transferred into a Falcon® 5 ml Round Bottom Polystyrene Test Tube, with Cell Strainer Snap Cap and subsequently FACS sorted.

## Cloning of IRF2BP1 and IRF2BP2 constructs

mRNA was extracted from HeLa cells using the NucleoSpin® RNA II kit (Macherey-Nagel) according to the manufacturer's instructions. Following this, mRNA was transcribed into cDNA using the Super-ScriptTM First-Strand Synthesis System (Invitrogen). IRF2BP1 and IRF2BP2 open reading frames were amplified by PCR using gene-specific primers, which also contained the restriction sites BamHI and XhoI and Phusion DNA Polymerase (Thermo Scientific). As a vector backbone, a modified pcDNA3.1 vector was used that already contained an N-terminal HA-tag. IRF2BP1 and IRF2BP2 PCR fragments were cloned into this vector using BamHI and XhoI restriction sites. Lysine to arginine mutations were introduced using the QuickChange® Site-directed Mutagenesis Kit (Stratagene) and site-specific mutagenesis primers.

For generation of bi-cistronic IRF2BP1/GFP constructs, PCRs were performed from HA-IRF2BP1 constructs using gene-specific primers, which also contained the restriction sites BamHI and EcoRI, and Phusion DNA Polymerase (Thermo Scientific). As a vector backbone, pIRES-hrGFPII was used. IRF2BP1 PCR fragments were cloned into this vector using BamHI and EcoRI restriction sites. Site-directed mutagenesis was done to generate constructs that are resistant to siRNA#5.

Sequencing of all constructs was done at GATC using a vector-specific CMV primer as well as internal sequencing primers.

## Protein gels and Western blotting

For most of the gels and blots, 6, 8, or 10% Schägger gels were used (Schägger & Jagow, 1987; Schägger, 2006)). Blotting was done onto nitrocellulose membranes (Protran, Amersham 0.45), using a buffer containing 25 mM Tris, 193 mM glycine, 20% ethanol, and 0.04% SDS.

After blocking membranes with 5% milk in PBS-T (1× PBS, 0.1% Tween20), primary antibodies were applied for at least 2 h and secondary, HRP-conjugated antibodies were applied for 1 h. Western blots were developed using enhanced chemiluminescent (ECL) horseradish peroxidase (HRP) substrate (SuperSignal™ West Pico PLUS from Thermo Scientific for standard signals or Immobilon Western HRP Substrate from Merck Millipore for weak signals), and the luminescent signals were detected using LAS 4000 biomolecular imager.

## Endogenous SUMO immunoprecipitations

Endogenous SUMO immunoprecipitations (SUMO-IPs) were performed as first described in (Becker *et al*, 2013). For that we grew hybridoma cells to generate antibodies against SUMO1 (clone 21C7, Developmental Studies Hybridoma Bank) and SUMO2/3 (Clone 8A2, Developmental Studies Hybridoma Bank) and used CELLine bioreactors to harvest antibodies at a concentration of approximately 1 mg/ml.

For generating SUMO antibody beads, 8 mg of SUMO antibody was added to 1 mg protein G agarose beads (Roche) for 1.5 h at 4°C. Beads were washed with 20 mM sodium phosphate buffer, pH 7.4, and crosslinked using 20 mM DMP in 50 mM sodium tetraborate, pH 9.0 for 1.5 h at room temperature. Afterward, beads were quenched in 50 mM Tris, pH 8.0, and washed in 20 mM sodium phosphate buffer, pH 7.4 before use.

To obtain cell lysates, 80% confluent HeLa cells growing adherently on 15 cm plates were lysed directly in 500 μl denaturing lysis buffer (1× PBS, 2% SDS, 10 mM EDTA, 10 mM EGTA, 2 mM Pefa-blocker, 2 μg/ml aprotinin, 2 μg/ml pepstatin, 2 μg/ml leupeptin, 20 mM NEM), leading to a final volume of 1 ml lysate (final SDS concentration 1%). The samples were snap frozen in liquid nitrogen and stored at −80°C.

The cell lysates were quickly thawed, supplemented with 75 mM DTT and boiled for 10 min at 60–70°C. The lysates were sonicated until the DNA sheared completely and diluted 1:10 in cold pre-RIPA dilution buffer (20 mM $NaH_2PO_4/Na_2HPO_4$, pH 7.4, 150 mM NaCl, 1% Triton X-100, 0.5% sodium deoxycholate, 5 mM EDTA, 5 mM EGTA, 1 mM Pefa-blocker, 1 μg/ml aprotinin, 1 μg/ml pepstatin, 1 μg/ml leupeptin, 20 mM NEM). Upon dilution, the final extract now corresponds to RIPA buffer. Finally, extracts were sterile filtered (0.45 μm pores) in order to obtain the input of the SUMO-IPs. Further steps were performed on ice if not stated otherwise.

The filtered input was added to the beads pre-equilibrated in the RIPA dilution buffer (100 μl beads/10 mg protein in the input sample, as determined by Bradford assay) and incubated overnight upon gentle rotation at 4°C. The beads were washed 3× in RIPA buffer (20 mM $NaH_2PO_4/Na_2HPO_4$, pH 7.4, 150 mM NaCl, 1%

Triton X-100, 0.5% sodium deoxycholate, 0.1% SDS, 5 mM EDTA, 5 mM EGTA, 1 mM Pefa-blocker, 1 μg/ml aprotinin, 1 μg/ml pepstatin, 1 μg/ml leupeptin, 10 mM NEM) in 1.5-ml LoBind tubes (Eppendorf). Next, the samples were incubated at 37°C with three volumes (300 μl per 100 μl beads) of the pre-elution buffer (RIPA buffer containing 500 mM NaCl, supplemented freshly with 10 mM NEM) to elute unspecifically bound proteins during gentle mixing. Elution of SUMOylated proteins was done for at least 30 min with three volumes of elution buffer (RIPA buffer containing 500 mM NaCl and 0.5 mg/ml SUMO epitope peptide) during gentle mixing. This elution step was done twice, and the eluates were harvested into fresh LoBind tubes. SUMOylated protein eluates were concentrated using TCA precipitation (10% TCA for 1 h at 4°C, followed by two washes in acetone at −20°C).

Samples were then loaded onto SDS–PAGE gels and analyzed by immunoblotting or mass spectrometry.

## Microarray

### RNA preparation

For comparing the RNA expression levels between IRF2BP1 wild type and the SUMO-deficient KR mutant, HeLa cells stably expressing the pIRES-hrGFPII based vectors (IRF2BP1 WT or IRF2BP1 KR, 4[th] FACS sorting) were treated with siRNA#5 against IRF2BP1 for 72 h in a six-well. After 56 h, cells were serum-starved for (a) 16 h and (b) 15 h and treated for 1 h with 100 ng/ml EGF or (c) not serum-starved at all.

For comparing the RNA expression levels between different knock-downs, six-well plates of HeLa cells were treated with siRNAs (non-targeting, IRF2BP1#5, IR2BP2#13) for 72 h in a six-well. After 56 h, cells were serum-starved for (a) 16 h or (b) 15 h and treated for 1 h with 100 ng/ml EGF.

Efficiency of the siRNA was controlled by using the siRNA transfection mixture on normal HeLa cells in parallel and by checking the protein amounts of IRF2BP1 in immunoblotting.

Each experiment was done in three independent experiments.

RNA samples were purified using Nucleospin RNA Plus kit (Macherey-Nagel) following manufacturer's instructions. RNA was tested by capillary electrophoresis on an Agilent 2100 bioanalyzer (Agilent), and high quality was confirmed.

Gene expression profiling was performed using GeneChip™ HuGene 2.0 ST Array (Affymetrix). Biotinylated antisense cRNA was then prepared according to the Affymetrix standard labeling protocol with the GeneChip® WT Plus Reagent Kit and the GeneChip® Hybridization, Wash and Stain Kit (both from Affymetrix). Afterward, the hybridization on the chip was performed on a GeneChip Hybridization oven 640, then dyed in the GeneChip Fluidics Station 450, and thereafter scanned with a GeneChip Scanner 3000. All of the equipment used was from Affymetrix (High Wycombe, UK).

### Bioinformatics

A Custom CDF version 21 with ENTREZ-based gene definitions was used to annotate the arrays (Dai *et al*, 2005). The Raw fluorescence intensity values were normalized applying quantile normalization and RMA background correction. Before performing the ANOVA, a batch normalization was used to remove the individual mouse variations. An ANOVA was performed to identify differential expressed

genes using a commercial software package SAS JMP Genomics, version 7, from SAS (SAS Institute, Cary, NC, USA). A false-positive rate of $a = 0.05$ with FDR correction was taken as the level of significance.

For Fig 3A, intensity values for each gene in the wild type and K579 mutant cell lines with and without EGF were measured in triplicates (arising from three biological replicates). Each value was transformed to its $\log_2$ value; then, an average was formed for each triplicate. The $\log_2$-fold change was then calculated by subtracting the value obtained for the −EGF from the value obtained for +EGF in both cell lines.

Gene Set Enrichment Analysis (GSEA) was used to determine whether defined lists (or sets) of genes exhibit a statistically significant bias in their distribution within a ranked gene list using the software GSEA (Subramanian et al, 2005). Pathways belonging to various cell functions such as cell cycle or apoptosis were obtained from public external databases (KEGG, http://www.genome.jp/kegg) and the Hallmark Geneset database.

The raw and normalized data are deposited in the Gene Expression Omnibus database (http://www.ncbi.nlm.nih.gov/geo/; accession numbers GSE135221 and GSE161716).

## Chromatin IPs

HeLa cells stably expressing pIRES-hrGFPII-based vectors (IRF2BP1 WT or IRF2BP1 KR, 4[th] or 7[th] FACS sorting) were treated with siRNA#5 against IRF2BP1 for 72 h on a 10-cm plate. After 56 h, cells were serum-starved for 16 h or for 15 h and treated for 1 h with 100 ng/ml EGF. Chromatin IP (ChIP) experiments were performed using the SimpleChIP enzymatic chromatin IP kit (Cell Signalling), exactly following manufacturer's instructions.

The chromatin digestion efficiency was systematically controlled, and the chromatin concentration was also monitored. The IP was performed with 10 µg of digested chromatin and 2 µg of IRF2BP1 or IRF2BP2 antibodies (Proteintech) or normal rabbit IgG provided in the kit.

DNA abundance was quantified by qPCR (see below).

## qPCR

RNA was purified using the Nucleospin RNA Plus kit (Macherey-Nagel). RNA concentration and quality was determined by Nano-Drop. Two microgram of RNA was used to produce cDNA using the High-Capacity cDNA Reverse Transcription Kit (Thermo Fisher, previously Applied Biosystem) following manufacturer instruction. qPCR reactions were prepared using the LightCycler® 480 SYBR Green kit (Roche) according to the manufacturer's instructions.

For checking DUSP1 expression on the RNA level, DUSP1 qPCR primers and ACTB primers were used. The qPCR was performed with the following cycle: denaturing at 95°C for 10 s, annealing at 56°C for 10 s, and extension at 72°C for 10 s. Technical triplicates were done in each biological experiment, and the data were analyzed using the ΔΔCt method.

For ChIP-qPCRs of DUSP1 and ATF3, we used 2 µl of sample (input was 1:5 diluted). Denaturing was done for 10 s at 95°C, annealing for 10 s at 56–59°C using the primers "TSS" (56°C), "TSS-67" (59°C), "TSS-224" (58°C), "TSS-961" (59°C) and "ATF-TSS-241" (54°C), and extension for 20 s at 72°C.

## EGFR surface staining

HeLa cells were serum-starved for 16 h (to remove all EGF at the plasma membrane that might interfere with EGFR binding) and afterward washed twice with PBS. They were then incubated with a solution containing PBS/2.5 mM EDTA/25mM HEPES/2% FBS (dialyzed at 12–14 MWCO) for 5 min at 37°C and then detached from cell culture plates using a cell lifter. Cells were centrifuged for 5 min at 300 $g$, resuspended in 1 ml PBS/2.5 mM EDTA, and counted. $10^6$ cells were used for each experiment and blocked with 2% FBS (dialyzed at 12–14 MWCO) for 20 min at 4°C (rotating). One µg anti-EGFR-Alexa Fluor® 555 antibody (neutralizing, clone LA1, Merck) was added to each $10^6$ cells for 1 h at 4°C (rotating). Cells were then washed three times in PBS and directly analyzed for Alexa 555 signal (using a 561 nm laser) using a BD FACSCanto™.

## Migration assays (IncuCyte®)

HeLa cells were reverse-transfected with siRNA against IRF2BP1 or non-targeting in one 12-well each. After 1 day, they were split and counted using LUNA™ cell counter and a 96-well plate was prepared with 10,000 cells per well (three wells were done as technical replicates for each biological experiment). One day after plating, a scratch wound was created using the IncuCyte® WoundMaker tool (Essenbioscience) and cells were immediately analyzed in an IncuCyte® (Essenbioscience) for 4 days, taking one image per well every 2 h. Relative wound density was determined using the manufacturer's program.

## Proliferation assays (IncuCyte®)

HeLa cells were reverse-transfected with siRNA against IRF2BP1, IRF2BP2, and non-targeting in one 12-well each. After 3 days, they were split and FACS sorted to obtain a defined amount of single cells. A 96-well plate was prepared with 5,000 cells per well (five wells were done as technical replicates for each biological experiment), and they were again reverse transfected with their respective siRNA. Cells were immediately analyzed in an IncuCyte® (Essenbioscience) for 6 days, taking four images per well every 2 h. Confluence of the cells was determined using the manufacturer's program.

## Quantification of the SUMO proteome upon EGF/mass spectrometry

HeLa cells were grown for 6–7 doublings (from 30 to 3,600 cm² culture dish surface, i.e. 24 15 cm plates) in SILAC DMEM medium containing dialyzed FBS (dialyzed 3× against PBS through a 6–8.000 MWCO bag), 2 mM L-glutamine, and 146 µg/ml lysine and 86 µg/ml arginine. One set of 24 plates contained "light" lysine and arginine, and the other set of 24 plates contained D4-lysine and $^{13}$C-arginine. 16–18 h before collecting the cells, they were serum-starved in SILAC DMEM medium containing only glutamine and the respective type of lysine and arginine.

One set of cells was treated with 100 ng/ml EGF (in PBS-BSA) for 10 min; the other set was treated with PBS-BSA only. For large-scale SUMO-IPs, the cells were lysed in 350 µl 2-× lysis buffer per

plate and lysates from all 48 plates were combined. SUMO-IPs were performed as stated above, and TCA precipitated eluates were loaded onto NuPAGE® Novex® Bis-Tris Mini Gels (4–12%) and stained with Coomassie Brilliant Blue.

In-gel digestion was performed as described (Shevchenko *et al*, 2006) with minor modifications. Unless otherwise stated, all incubation steps were performed at 26°C in an Eppendorf thermomixer at 1,050 rpm for 15 min and all solutions were prepared with LiChrosolv $H_2O$. Each lane of the stained gel was cut into 23 equally sized pieces. Gel slices were first washed with 150 µl $H_2O$ and dehydrated with 150 µl acetonitrile (ACN). Dried gel pieces were incubated with 150 µl of 10 mM DTT reducing solution for 50 min and then alkylated with 55 mM iodoacetamide for 20 min at 26°C in the dark. After another round of dehydration of gel pieces with ACN, they were rehydrated with 60 µl of digestion buffer (2 µg/ml trypsin in 25 mM $NH_4HCO_3$, pH 8.5) and incubated at 37°C over night.

For peptide extraction, supernatants of the three following steps were pooled: First, 100 µl ACN were added to dehydrate the gel pieces. Second, gel pieces were rehydrated with 50 µl 5% [v/v] formic acid (FA), followed by addition of 50 µl ACN. Third, 50 µl ACN were added to fully dehydrate the gel pieces. The three supernatants containing the peptides were pooled and dried in vacuum centrifuge (Thermo Scientific). Dried peptides were stored at −20°C until submitted to LC-MS.

The extracted peptides initially were dissolved in 20 µl 3% ACN/ 1% [v/v] FA by vortexing and brief sonication on water bath. For each MS run, 5 µl was loaded onto an in-house packed C18 trap column. Retained peptides were eluted and separated on an analytical C18 capillary column at a flow rate of 300 nl/min with a gradient from 5 to 37% acetonitrile in 0.1% formic acid for 50 min including column equilibrium and wash by using an Agilent 1100 nano-flow LC system (Agilent Technologies, Santa Clara, CA). Agilent 1100 nano-flow LC was coupled to LTQ-Orbitrap XL (Thermo Electron, Bremen, Germany), and it was operated in a data-dependent mode. The survey scans were acquired in the Orbitrap ($m/z$ 350–1,600) with a resolution of 30,000 at $m/z$ 400 with a target value of $10^6$. For up to five of the most intense ions with charges $\leq 2$ from the survey scan were sequentially selected for collision-induced dissociation (CID) in the LTQ linear ion trap with a normalized collision energy of 35%.

MaxQuant software and the Mascot search engine were used for analysis of raw MS files from the LTQ-Orbitrap XL. Quant.exe module of MaxQuant generated the peak lists were searched against the International Protein Index human protein database common contaminants (e.g. keratins, serum albumin) and concatenated with the reverse sequences of all entries. Database (Mascot) search parameters were set as: Cysteine carbamidomethylation was as a fixed modification, whereas methionine oxidation and N-terminal protein acetylation were as variable modifications; tryptic specificity with no proline restriction and up to two missed cleavages was set. The MS survey scans and MS/MS mass tolerance were set 7 ppm and 0.5 Da, respectively. A minimal length of six amino acids was considered for identification. The false discovery rate was set to 1% at both the peptide and the protein level. For identification and quantification, a posterior error probability (PEP) of peptides was required to be at maximum 0.05. Re-quantify was enabled, and "keep low scoring versions of identified peptides" was disabled. A minimum ratio count of one for each protein was required for quantification of SILAC pairs by considering unique and razor peptides.

In total, this large-scale SILAC/SUMO-IP/Mass spectrometry experiment was done three times, once where "no EGF" was labeled "light" and "10-min EGF treatment" was labeled "heavy", and twice vice versa. We used very stringent criteria to obtain the final list of hits: First, all contaminants and reverse sequences, as well as proteins with a "SigB" (calculated by MaxQuant) > 0.05, were removed. Second, only proteins that had a ratio count of 4 or higher were used. Third, proteins that were only identified in one of the tree experiments, or that showed a different behavior in one of the experiments, were removed. Fourth, if a protein was identified in two of the three experiments, it was neglected if the ratio difference was larger than 10-fold. Fifth, a ratio variability larger than 80 percent was not allowed.

## Immunofluorescence

24–48 h after siRNA transfection cells were seeded onto glass coverslips. Cells were fixed 4% paraformaldehyde (PFA) in PBS, washed twice with PBS, and permeabilized for 15 min in 0.1% Triton X-100. Cells were blocked for 60 min in blocking buffer (1% BSA in 0.1% Triton X-100). Afterward, cells were incubated for primary antibodies (1:200 for IRF2BP1 antibody, Proteintech) in a wet chamber for 2 h. After washing, primary antibodies were labeled with Alexa-conjugated secondary antibodies for 1 h in the dark in a wet chamber. Hoechst was included with the secondary antibody to stain DNA. All antibodies were diluted in blocking buffer. Images were taken using Axio Observer Z1 fluorescence microscope (Zeiss) equipped with an AxioCam MRm camera and a Plan-APOCHROMAT 63×/1.4 Oil DIC objective. Background subtraction and image brightness and contrast were adjusted equally using ImageJ.

## Chromatin binding

HeLa cells were harvested, and the cell pellet was lysed with 5× pellet volume (PV) of ice-cold cytoplasmic extract (CE) buffer (10 mM HEPES, 1 mM EDTA, 1 mM EGTA, 60 mM KCl, 1 mM Pefa, 0.075% NP40, and 1 µg/ml each of aprotinin, leupeptin and pepstatin, pH 7.6) and incubated on ice for 3 min. Cell lysis was controlled by trypan blue staining for an efficiency of at least 70%. The sample was centrifuged at 1,500 *g* for 4 min at 4°C, and the cytoplasmic extract (CE) was transferred to a fresh tube. The nuclear pellet was gently washed with 3xPV ice-cold wash buffer (CE buffer without NP40) by pipetting up and down and centrifuged at 1,500 *g* for 4 min at 4°C. The nuclear pellet was incubated with 2xPV ice-cold nuclear extract (NE) buffer (varying concentrations of NaCl, 20 mM HEPES, 1.5 mM $MgCl_2$, 0.2 mM EGTA, 25% glycerol, 1 mM Pefa, and 1 µg/ml each of aprotinin, leupeptin, and pepstatin, pH 7.9) and incubated on ice for 10 min, and the sample was vortexed periodically to re-suspend the pellet. Then, the sample was centrifuged at 1,500 *g* for 4 min at 4°C and the nuclear extract was transferred to a fresh tube. This step was done first with 170 mM NaCl ($NE_{170\ mM}$ buffer) and repeated with the new nuclear pellet and $NE_{290\ mM}$ and $NE_{420\ mM}$ buffers. The cytoplasmic and nuclear extracts were clarified by centrifugation at 20,000 *g* for 30 min at 4°C.

## Data availability

The datasets produced in this study are available in the following databases:

-Mass Spectrometry Data (SUMO-IP-SILAC, Dataset EV1): PRIDE PXD018049 (https://www.ebi.ac.uk/pride/archive/projects/PXD018049)
-Microarray data (IRF2BP1 WT and K579R cells, Dataset EV2): Gene Expression Omnibus GSE135221 (https://www.ncbi.nlm.nih.gov/geo/query/acc.cgi?acc=GSE135221)
-Microarray data (IRF2BP1 knockdown, Dataset EV3): Gene Expression Omnibus GSE161716 (https://www.ncbi.nlm.nih.gov/geo/query/acc.cgi?acc=GSE161716)

**Expanded View** for this article is available online.

## Acknowledgements

We thank Dr. Simona Polo for providing HeLa cells, antibodies, and ideas and gratefully acknowledge Heidi Ehret, Andrea Frank, Anja Schubert, and Ulrike Gern for excellent technical assistance. We thank Dr. Annette Flotho for critical reading of the manuscript and all members of the Melchior lab, Dr. Nils Blüthgen (Charité, Berlin) and Dr. Gianluca Sigismondo (DKFZ, Heidelberg) for help and useful discussions. This work received funding from the Deutsche Forschungsgemeinschaft (DFG, German Research Foundation) - Project Number 278001972 - TRR 186, the DGF-funded Cluster of Excellence CellNetworks Postdoc Program (to SVB) and the Peter and Traudl Engelhorn Foundation (to SVB). Open Access funding enabled and organized by ProjektDEAL.

## Author contributions

SVB, NS-V, and FM conceptualized the study. SVB, NS-V, TM, SK, CS, and HU involved in methodology. SVB, NS-V, TM, SK, JD, TNA, AV, CS, and LW involved in investigation. SVB and FM wrote the original draft. SVB, NS-V, TM, and FM wrote, reviewed, and edited the manuscript. SVB and FM involved in funding acquisition. FM and HU involved in supervision.

## Conflict of interest

The authors declare that they have no conflict of interest.

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
