## [Review Process File · EMBO Reports]

Transient deSUMOylation of IRF2BP proteins controls early transcription in EGFR signaling

Sina Barysch, Nicolas Stankovic-Valentin, Tim Miedema, Samir Karaca, Judith Doppel, Thiziri Nait Achour, Aarushi Vasudeva, Lucie Wolf, Carsten Sticht, Henning Urlaub, and Frauke Melchior

DOI: [10.15252/embr.201949651](https://doi.org/10.15252/embr.201949651)

Corresponding author(s): [Frauke Melchior \(f.melchior@zmbh.uni-heidelberg.de\)](mailto:f.melchior@zmbh.uni-heidelberg.de)

Review Timeline:

Submission Date:	12th Nov 19
Editorial Decision:	23rd Dec 19
Revision Received:	22nd Nov 20
Editorial Decision:	9th Dec 20
Revision Received:	16th Dec 20
Accepted:	21st Dec 20

Editor: *Martina Rembold*

Transaction Report:

Dear Frauke,

Thank you for the submission of your research manuscript to our journal. We have now received the full set of referee reports that is copied below.

As you will see, the referees acknowledge that the findings are potentially interesting but they also have a number of suggestions on how to strengthen the dataset. Referee 1 suggests testing the specificity of K579 SUMOylation with SUMO E1 inhibitors and both, referee 1 and 2 suggest to further test the proposed effect on MAPK signalling. Referee 3 has several suggestions to dissect how SUMOylated IRF2BP affects DUSP expression. While a detailed mechanistic understanding will not be required, these experiments would certainly further strengthen the manuscript.

Given these constructive comments, we would like to invite you to revise your manuscript with the understanding that the referee concerns (as detailed above and in their reports) must be fully addressed and their suggestions taken on board. Please address all referee concerns in a complete point-by-point response. Acceptance of the manuscript will depend on a positive outcome of a second round of review. It is EMBO reports policy to allow a single round of revision only and acceptance or rejection of the manuscript will therefore depend on the completeness of your responses included in the next, final version of the manuscript.

Revised manuscripts should be submitted within three months of a request for revision; they will otherwise be treated as new submissions. Please contact us if a 3-months time frame is not sufficient for the revisions so that we can discuss the revisions further.

- 1) A data availability section providing access to data deposited in public databases is missing (if relevant).
- 2) Your manuscript contains error bars based on $n=2$. Please use scatter blots showing the individual datapoints in these cases. The use of statistical tests needs to be justified.

2) individual production quality figure files as .eps, .tif, .jpg (one file per figure).

Please download our Figure Preparation Guidelines (figure preparation pdf) from our Author Guidelines pages

<https://www.embopress.org/page/journal/14693178/authorguide> for more info on how to prepare your figures.

3) a .docx formatted letter INCLUDING the reviewers' reports and your detailed point-by-point

responses to their comments. As part of the EMBO Press transparent editorial process, the point-by-point response is part of the Review Process File (RPF), which will be published alongside your paper.

4) a complete author checklist, which you can download from our author guidelines (). Please insert information in the checklist that is also reflected in the manuscript. The completed author checklist will also be part of the RPF.

5) Please note that all corresponding authors are required to supply an ORCID ID for their name upon submission of a revised manuscript (). Please find instructions on how to link your ORCID ID to your account in our manuscript tracking system in our Author guidelines ()

6) We replaced Supplementary Information with Expanded View (EV) Figures and Tables that are collapsible/expandable online. A maximum of 5 EV Figures can be typeset. EV Figures should be cited as 'Figure EV1, Figure EV2" etc... in the text and their respective legends should be included in the main text after the legends of regular figures.

7) Data Availability section: Please follow the model below to list the accession number for the microarray dataset.

The accession numbers and database should be listed in a formal "Data Availability " section (placed after Materials & Method) that follows the model below (see also <<https://www.embopress.org/page/journal/14693178/authorguide#dataavailability>>). Please note that the Data Availability Section is restricted to new primary data that are part of this study.

Data availability

8) We would also encourage you to include the source data for figure panels that show essential data. Numerical data should be provided as individual .xls or .csv files (including a tab describing the data). For blots or microscopy, uncropped images should be submitted (using a zip archive if multiple images need to be supplied for one panel). Additional information on source data and instruction on how to label the files are available .

10) Regarding data quantification:

- Please ensure to specify the name of the statistical test used to generate error bars and P values, the number (n) of independent experiments underlying each data point (not replicate measures of one sample), and the test used to calculate p-values in each figure legend. Discussion of statistical methodology can be reported in the materials and methods section, but figure legends should contain a basic description of n, P and the test applied.

IMPORTANT: Please note that error bars and statistical comparisons may only be applied to data obtained from at least three independent biological replicates. If the data rely on a smaller number of replicates, scatter blots showing individual data points are recommended and statistical tests must be justified..

- Graphs must include a description of the bars and the error bars (s.d., s.e.m.).

11) As part of the EMBO publication's Transparent Editorial Process, EMBO reports publishes online a Review Process File to accompany accepted manuscripts. This File will be published in conjunction with your paper and will include the referee reports, your point-by-point response and all pertinent correspondence relating to the manuscript.

I look forward to seeing a revised version of your manuscript when it is ready. Please let me know if you have questions or comments regarding the revision.

Kind regards and merry Christmas,
Martina

Martina Rembold, PhD
Editor
EMBO reports

Referee #1:

This is a highly interesting study from the group of SUMO pioneer Frauke Melchior on SUMOylation dynamics in response to EGF receptor signalling using a SILAC-based quantitative mass-spectrometry approach. I am enthusiastic about this work and have several comments as detailed below.

Major comments:

1-Lysines are subject to multiple different types of post-translational modifications (PTMs). Lysine to arginine mutations could thus also interfere with other PTMs. To rule out that the observed phenotype is due to interfering with other PTMs, I strongly recommend a control experiment using commercially available ML-792 as a specific SUMO E1 inhibitor that can disrupt sumoylation of the wild-type proteins and possibly reveal a more striking phenotype due to functional protein group regulation. An alternative control that is recommended is to make a E581A mutant of IRF2BP1 that would equally well disrupt sumoylation, leaving lysine K579 intact for other PTMs.

2-The mechanism for the surprisingly transient desumoylation of TRIM24, IRF2BP1 and IRF2BP2 is unclear. The authors speculate that an upstream transient signalling event may be involved and they consider phosphorylation of IRF2BP1 or one of its binding partners, or phosphorylation of a SUMO enzyme as alternatives. Since the SUMO system contains only very few enzymes and these enzymes consequently have larger sets of target proteins, the latter scenario is unlikely. It would be good to include this in the discussion section. Furthermore, phosphorylation has extensively been reported to drive sumoylation, therefore a transient dephosphorylation of TRIM24, IRF2BP1 and IRF2BP2 seems more likely to result in a transient desumoylation of these proteins. A wealth of information on phosphorylation and ubiquitylation dynamics in response to EGF is available from large-scale mass spectrometry studies. The authors could investigate this information to share in the discussion what is already known on this topic for relevant proteins.

3-In the model, DUSP1 is shown to dephosphorylate MAPK. The authors need to investigate whether this difference in MAPK dephosphorylation can be observed in the lysates used for Figure 3G.

4-Sharing the underlying raw and processed mass spectrometry data via the PRoteomics IDentifications (PRIDE) database is an excellent practice in the field and is strongly recommended.

Minor comments:

5-Page 5. "charge swapping" should be "label swapping".

6-The microscopy data in Figure 2F is hardly visible.

Referee #2:

The manuscript by Barysch et. al. describes a novel and interesting regulatory loop involving the EGF-dependent SUMOylation/deSUMOylation of the transcriptional co-regulators IRF2BP1/2, in turn controlling downstream EGFR signaling. In particular, the authors showed that the deSUMOylation of IRF2BP1 induced by EGF treatment upregulated the transcription of the phosphatase DUSP-1, a known regulator of MAPK signaling. The author proposed that SUMO could serve as an important brake to prevent an overshoot and uncontrolled response to EGF. The message underlined the manuscript is novel and of broad interest for the cell biology and signaling community. Experiments are technically well performed and carefully controlled. The workflow of the manuscript is clear and logic.

Only one point should be further dissected, in my opinion, in order to extend the biological relevance of the observed regulatory mechanism.

How the SUMOylation-dependent transcriptional effects on DUSP1 (and, possibly, on other EGF-dependent transcriptional targets) are decoded by the cells in terms of alterations in EGFR signaling and cellular response?

For instance, it would be important to show if there is an impact of the RF2BP1 SUMOylation-defective mutant on MAPK signaling activation response (or other EGF-dependent signaling). In addition, showing whether the altered balance between SUMOylation/deSUMOylation of IRF2BP1 has an impact on EGF-induced proliferation (e.g. BrdU incorporation) or migration (e.g. transwell or wound-healing assay) would strengthen the message of the manuscript.

Referee #3:

In this report Barysch and coworkers address the question how the cellular SUMO proteome is changed in response to EGF stimulation. In a system-wide proteomics approach they observed a transient deSUMOylation of a small number of transcription factors. Among these the transcriptional repressors IRF2BP1, IRB2BP2 and IRF2BP2L were transiently demodified at an early time point after EGF stimulation of HeLa cells. To define the functional consequence of this process the authors mapped the SUMO attachment sites in IRF2BP1/2 and generated cell lines stably expressing wild-type or a SUMO-deficient variant of IRF2BP1 (IRF2BP1 K579R). Loss of SUMO modification did not affect stability, localization or chromatin binding of IRF2BP1. Interestingly, however, a subset of EGF-responsive genes exhibit a differential expression pattern when comparing cells with wild-type versus SUMO-deficient IRF2BP1. To validate these findings the authors concentrated on expression of DUSP1, an established feedback regulator of EGF signaling. Using CHIP and qPCR experiment they demonstrate that IRF2BP1 binds to the DUSP1 promoter in proximity to the TSS and limit DUSP1 expression in a SUMOylation-dependent process. Based on these data it is proposed that SUMOylation of IRF2BP1/2 functions as a molecular switch in controlling EGF signaling.

Altogether this is an interesting study, which connects EGF signaling to the SUMO pathway. The experimental data are of high quality and for the most part very convincing. Performing comparative SUMO proteomics on endogenously SUMOylated proteins is a major strength of this work. Moreover, the model of a molecular switch through transient SUMO conjugation/deconjugation of a transcriptional regulators is an attractive concept. A considerable drawback of the manuscript is that it currently does not provide mechanistic insight how the transient desumoylation of IRF2BP is triggered and how SUMOylated IRF2BP limits expression of DUSP1. I appreciate that it is quite

challenging to address these points, but I would suggest that prior to publication the authors do at least some candidate-based experiments to better define the SUMO-dependent repressive effect.

Major point:

1) As outlined above the manuscript lacks mechanistic data on the SUMO-dependent repression of DUSP1. SUMO-dependent repression of gene expression is a common concept and at least in some cases it has been demonstrated that it relies on SUMO-dependent promoter recruitment of co-repressors, including NCOR-SMRT-HDAC containing complexes. Since IRF2BP2 was found to be associated with NCOR, Sin3A and HDAC (see for example BioGRID database) it would be interesting to see whether any of these co-repressors is recruited to the DUSP1 promoter in a SUMO-dependent manner. This could be addressed by CHIP experiments in IRF2BP1 wild-type versus IRF2BP1 K579R expressing cells.

2) The authors propose that IRF2BPs are rapidly deSUMOylated upon EGF stimulation. Could the involvement of distinct SUMO deconjugases in this process be tested by knock-down experiments? If feasible a distinct deconjugase might be identified and its differential interaction with IRF2BPs upon EGF treatment could be tested. This could strengthen the switch model, but would indeed need considerable experimental input.

Minor points:

1) According to the methods section the microarray experiment was done in triplicate for each cell-line, but Figure 3A does not show any error bars. Is this Figure based on only one selected dataset? One would rather like to see the mean of all three experiments with error bars.

Response to referees

Dear Reviewers,

Thank you very much for your interest in our study, your very positive evaluation and your insightful comments and suggestions. Our sincere apologies for the long time it took until resubmission. This is not due to fundamental problems that we had with the revision, but after initial problems with discontinued antibodies (anti Dusp1 from Santa Cruz), we were first hit by Corona restrictions and lockdowns, but in addition, our institute "collapsed" due to a water pipe breakage, which destroyed the complete electricity system of the ZMBH. It took three months to relocate most of the ZMBH researchers (including us) into some emergency space on campus, where first experiments could only be started again in October.

Nevertheless, we have finally been able to carry out many new experiments and are excited to finally share them with you. We sincerely hope to have addressed your comments in a satisfactory way. Some experiments are now included in the revised manuscript, others are shown for your attention below.

Overview of the Figures in the revised manuscript:

- a) Figure 1: unchanged
- a) Figure 2F: the immunofluorescence panel has been replaced for better visibility
- b) Figure 3:
 - the old panel 3G is now the new panel 4A,
 - panel 3F (stable cell lines, Dusp1 immunoblot) has been removed (see answer to reviewer 2)
 - New panel 3F shows DUSP1 qPCR after EGF for a new cell line, IRF2BP1 V578A
 - New panel 3G shows qPCR after EGF for three additional genes, Fos, ATF3, Egr2
- c) Figure 4: had only one panel (a model) in the first submission.
 - New 4A is the old panel 3G with one additional panel (immunoblotting for ATF3)
 - New 4B is the model, which has been slightly modified
 - New 4C - 4H are new data, addressing migration, proliferation, gene expression upon knockdown of IRF2BP1 (and in parts also IRF2BP2).

Overview of the Extended View Figures and Tables:

- a) Figure EV1: unchanged
- b) Figure EV2: the old figure, which had visualized public data, IRF2BP2 and PolIII on the mouse DUSP1 promoter, has been removed. We replaced it with a new Figure that confirms IRF2BP1 on the human ATF3 promoter by CHIP/PCR.
- c) Figure EV3: this is a new figure showing Gene Set Enrichment Analyses for the new microarray data.
- d) Figure EV4: this is the old, but slightly modified, model Figure EV3.
- e) Dataset EV1: unchanged
- f) Dataset EV2: unchanged
- g) Dataset EV3: new Microarray data, IRF2BP1 knockdown
- h) Table EV1: this 8 page table lists all materials used in this study.

We will of course share all proteomics and microarray data via appropriate outlets.

The microarray data from the stable cell lines are accessible via the GEO Series accession number GSE135221

The new microarray data are not yet public, but accessible for you here:
Gene Expression Omnibus, GSE161716 (Microarray kd) {Token removed}

The proteome data are available in EV Table X and via the ProteomeXchange server Identifier PXD018049

Step by Step response to the reviewers:

Blue and cursive are the reviewer's statements, black and regular are our answers.

Referee #1:

This is a highly interesting study from the group of SUMO pioneer Frauke Melchior on SUMOylation dynamics in response to EGF receptor signalling using a SILAC-based quantitative mass-spectrometry approach. I am enthusiastic about this work and have several comments as detailed below.

Thank you very much!

We have addressed your comments as follows:

Major comments:

1) Lysines are subject to multiple different types of post-translational modifications (PTMs). Lysine to arginine mutations could thus also interfere with other PTMs. To rule out that the observed phenotype is due to interfering with other PTMs, I strongly recommend a control experiment using commercially available ML-792 as a specific SUMO E1 inhibitor that can disrupt sumoylation of the wild-type proteins and possibly reveal a more striking phenotype due to functional protein group regulation. An alternative control that is recommended is to make a E581A mutant of IRF2BP1 that would equally well disrupt sumoylation, leaving lysine K579 intact for other PTMs.

You are of course right, and we therefore followed both of your suggestions to further validate a role for SUMO rather than other modifications.

We started with ML-792 as it was the easier experiment and a very nice idea. However, as seen in several independent experiments, ML-792 inhibited rather than increased DUSP1 transcription.

Figure 1 for reviewer's attention. The SUMO E1 inhibitor ML-792 reduces DUSP1 expression. Pilot experiments revealed that 1 hour treatment with 0.8 microM ML-792 is sufficient to abolish IRF2BP1 SUMOylation. HeLa cells were serum starved, incubated for 1 hour with DMSO or 0.8 microM ML-792 in DMSO, and subsequently treated with 100 ng/ml EGF for the indicated times. DUSP1 mRNA was quantified by qPCR, DUSP1 protein by immunoblotting (qPCR n=2, protein n=3).

This result is the opposite of what we expected from our IRF2BP1 K578R cell line, where transcription is enhanced in mutant cells. This could either be because the K578R mutation did more than abolish IRF2BP SUMOylation, or - and this is what we consider much more likely for reasons outlined below - because one hour treatment with ML-792 will inhibit SUMOylation of all proteins that go through reasonably fast cycles of modification and demodification, not just the three IRF2BP proteins. Amongst the >1000 known SUMOylated proteins in HeLa cells are many that contribute to constitutive and regulated transcription. We assume some of those will need SUMO for the machinery to properly function. It will be interesting to follow up on, but this is clearly beyond the scope of the manuscript.

We then followed the second suggestion, and generated a different cell line that expresses another SUMOylation deficient IRF2BP1. For this, we decided to generate cells expressing IRF2BP1 V578A. This mutation is less drastic than E581A, but it is nevertheless sufficient to abolish SUMOylation. Consistent with our findings for K579 IRF2BP1 cells, DUSP1 transcription upon EGF treatment was increased in IRF2BP1 V578A cells compared to HeLa cells expressing wt IRF2BP1.

This significantly strengthens our interpretation that the observed increase of DUSP1 transcription in the IRF2BP1 K578R cell line is caused by deficiency in SUMOylation and not due to other lysine modifications or folding problems, and we show the new data in the revised Figure 3F.

2) The mechanism for the surprisingly transient desumoylation of TRIM24, IRF2BP1 and IRF2BP2 is unclear. The authors speculate that an upstream transient signalling event may be involved and they consider phosphorylation of IRF2BP1 or one of its binding partners, or phosphorylation of a SUMO enzyme as alternatives. Since the SUMO system contains only very few enzymes and these enzymes consequently have larger sets of target proteins, the latter scenario is unlikely. It would be good to include this in the discussion section.

We agree with this reviewer, and added a short paragraph in the discussion:

“Either model depends on a signaling event that will inhibit SUMOylation, or stimulate deSUMOylation, of IRF2BP1. Very few sumoylated proteins lose SUMO in response to EGF - we consider it therefore likely that IRF2BP1 itself, rather than one of the SUMO enzymes, is altered.”

Furthermore, phosphorylation has extensively been reported to drive sumoylation, therefore a transient dephosphorylation of TRIM24, IRF2BP1 and IRF2BP2 seems more likely to result in a transient desumoylation of these proteins. A wealth of information on phosphorylation and ubiquitylation dynamics in response to EGF is available from large-scale mass spectrometry studies. The authors could investigate this information to share in the discussion what is already known on this topic for relevant proteins.

You are of course right that dephosphorylation is an option as well. And we changed the text in the discussion accordingly (“IRF2BP1 carries numerous phosphorylation sites, none of which is close to the SUMOylation consensus site. Whether (de)phosphorylation of any of these is responsible for the EGF-dependent SUMO switch awaits further investigation.”) Of course we have investigated available phosphorylation databases, and compendia (e.g., Ünal et al (2017) A compendium of ERK targets. FEBS Lett. 591, 2607-2615). And there is indeed also one candidate SP phosphorylation site in IRF2BP1 that seems to disappear upon EGF stimulation, S436 (Stuart et al 2015; Sharma et al 2014) , and we will follow up on it. But it is a long shot, requires a lot of work, and we rather discuss it once we have some evidence.

We kindly disagree that dephosphorylation is necessarily the more likely event. There are many good examples where phosphorylation of a specific target prevents recruitment of a PIAS E3 ligase, or where phosphorylation of a specific target allows recruitment of an isopeptidase (for examples, please see our 2013 Annual Review).

3) In the model, DUSP1 is shown to dephosphorylate MAPK. The authors need to investigate whether this difference in MAPK dephosphorylation can be observed in the lysates used for Figure 3G.

We thank the referee for this comment and we have analyzed MAPK dephosphorylation in our experimental setup. Unfortunately, the lysates of the experiment depicted in the manuscript were no longer available, we therefore repeated the experiment and probed these lysates for MAPK dephosphorylation as suggested by the referee. Although we see a consistent increase in DUSP1 expression in IRF2BP1 knock-down cells, this does not translate into changes in MAPK phosphorylation that we can detect by immunoblotting.

Figure 2 for reviewer's attention. IRF2BP1 knockdown increases DUSP1 expression upon EGF treatment, but does not accelerate MAPK dephosphorylation. HeLa cells were starved for 16 hours and subsequently treated with 100 ng/ml EGF for the indicated times. Cells were lysed and lysates were analyzed by immunoblotting with the indicated antibodies. * = unspecific band.

Why this is, we can only speculate about: DUSP1 itself is highly regulated by posttranslational modifications that influence its activity and stability, IRF2BP1 knockdown affects many other genes whose altered expression may influence MAPK phosphorylation, and DUSP1 and MAPK may not colocalize at this time.

To avoid overinterpretation, we decided to remove the feedback idea from our model figure. Instead we emphasize in the model, that several immediate early genes are affected by the SUMO switch.

4) Sharing the underlying raw and processed mass spectrometry data via the PRoteomics IDentifications (PRIDE) database is an excellent practice in the field and is strongly recommended.

Of course we will share the data. The ProteomeXchange identifier for our data is *PXD018049* and we have updated the information in our supplemental material. The data are not yet publicly available, [Reviewer access]

Minor comments:

5) Page 5. "charge swapping" should be "label swapping".

Thank you, we changed it in the text.

6) The microscopy data in Figure 2F is hardly visible.

Thanks for the comment. We agree and have replaced Figure 2F with better images.

Referee #2:

The manuscript by Barysch et. al. describes a novel and interesting regulatory loop involving the EGF-dependent SUMOylation/deSUMOylation of the transcriptional co-regulators IRF2BP1/2, in turn controlling downstream EGFR signaling. In particular, the authors showed that the deSUMOylation of IRF2BP1 induced by EGF treatment upregulated the transcription of the phosphatase DUSP-1, a known regulator of MAPK signaling. The author proposed that SUMO could serve as an important brake to prevent an overshoot and uncontrolled response to EGF.

The message underlined the manuscript is novel and of broad interest for the cell biology and signaling community. Experiments are technically well performed and carefully controlled. The work-flow of the manuscript is clear and logic.

We thank the referee for this kind assessment.

Only one point should be further dissected, in my opinion, in order to extend the biological relevance of the observed regulatory mechanism.

How the SUMOylation-dependent transcriptional effects on DUSP1 (and, possibly, on other EGF-dependent transcriptional targets) are decoded by the cells in terms of alterations in EGFR signaling and cellular response?

An important addition to our revised manuscript is the validation of three additional EGF-dependent transcriptional targets. We show in the revised Figure 3 that the immediate early genes ATF3, Egr2 and Fos also show increased transcription in response to EGF.

For instance, it would be important to show if there is an impact of the IRF2BP1 SUMOylation-defective mutant on MAPK signaling activation response (or other EGF-dependent signaling).

As described above, we could not detect a significant change of phosphoMAPK by immunoblotting, even when we depleted IRF2BP1, which gave the strongest induction of DUSP1. We have thus changed our model figure (now in Figure 4B) and discussion a bit to remove our speculation about DUSP1-dependent feedback regulation and to emphasize our finding that diverse Immediate Early Genes are affected by IRF2BP1 deSUMOylation.

In addition, showing whether the altered balance between SUMOylation/deSUMOylation of IRF2BP1 has an impact on EGF-induced proliferation (e.g. BrdU incorporation) or migration (e.g. transwell or wound-healing assay) would strengthen the message of the manuscript.

Even though the transcriptional changes in our two IRF2BP1 mutant cell lines are significant and highly reproducible, they are quantitatively not particularly strong. And they have been too small for us to detect convincing and reproducible phenotypes (we looked at phosphoMAPK, migration, proliferation and clonogenic survival).

Please note, we also had to realize that we could not consistently reproduce the small increase of DUSP1 protein in IRF2BP1 K579A cells that we had shown by immunoblotting in the original submission (old Figure 3F). Where the variability comes from, we have not been able to fully solve. In addition to technical issues (e.g. 20% differences are difficult to detect reliably by immunoblotting, variable knockdown efficiency increases variability, cells are rather unstable

and change over several weeks), Moreover, DUSP1 is strongly regulated, not only at the level of transcription, but also at the level of mRNA stability, by numerous phosphorylations and by very rapid degradation.

However, loss of a single SUMO is also not the physiological response to EGF - it is the coordinated loss of SUMO in all three IRF2BP proteins, which is likely to give a much more robust phenotype. Once we have identified the upstream signal, we will hopefully be able to address this.

To add to the significance of our findings in some other way, we decided to investigate the consequences of IRF2BP1 depletion. As we had already shown in the initial submission, depletion of IRF2BP1 prior to serum starvation and EGF treatment does lead to robust induction of DUSP1 protein. Here we added evidence that it also leads to significant induction of ATF3 (old Figure 3G, new Figure 4A, with new panel for ATF3).

We first investigated whether IRF2BP1 depletion in HeLa cells alters wound healing and cell proliferation. This is indeed the case (new data in Figure 4C and D). We then carried out microarray analyses to compare the transcriptome of HeLa cells with those in which we depleted IRF2BP1, IRF2BP2 or both (Figure 4E). Geneset enrichment analyses revealed a striking enrichment of genes linked to EGF receptor signaling (Figure 4F), and EGF receptor itself and its transport factor Sec24D were highly upregulated (4 fold on mRNA levels). We validated these findings by immunoblotting (Figure 4G) and we could show significantly elevated EGF receptor surface expression (Figure 4H). Together, these data reveal an unsuspected role for the poorly studied transcriptional regulator IRF2BP1 in the control of EGF receptor signaling. There is no evidence for direct binding of IRF2BP1 to the EGF receptor gene, and it is plausible to assume that the striking effect on EGFR has to do with IRF2BP1's role in immediate early gene expression.

Referee #3:

In this report Barysch and coworkers address the question how the cellular SUMO proteome is changed in response to EGF stimulation. In a system-wide proteomics approach they observed a transient deSUMOylation of a small number of transcription factors. Among these the transcriptional repressors IRF2BP1, IRB2BP2 and IRF2BP2L were transiently demodified at an early time point after EGF stimulation of HeLa cells. To define the functional consequence of this process the authors mapped the SUMO attachment sites in IRF2BP1/2 and generated cell lines stably expressing wild-type or a SUMO-deficient variant of IRF2BP1 (IRF2BP1 K579R). Loss of SUMO modification did not affect stability, localization or chromatin binding of IRF2BP1. Interestingly, however, a subset of EGF-responsive genes exhibit a differential expression pattern when comparing cells with wild-type versus SUMO-deficient IRF2BP1. To validate these findings the authors concentrated on expression of DUSP1, an established feedback regulator of EGF signaling. Using CHIP and qPCR experiment they demonstrate that IRF2BP1 binds to the DUSP1 promoter in proximity to the TSS and limit

DUSP1 expression in a SUMOylation-dependent process. Based on these data it is proposed that SUMOylation of IRF2BP1/2 functions as a molecular switch in controlling EGF signaling. Altogether this is an interesting study, which connects EGF signaling to the SUMO pathway. The experimental data are of high quality and for the most part very convincing. Performing comparative SUMO proteomics on endogenously SUMOylated proteins is a major strength of this work. Moreover, the model of a molecular switch through transient SUMO conjugation/deconjugation of a transcriptional regulators is an attractive concept.

Thank you very much for this positive feedback.

A considerable drawback of the manuscript is that it currently does not provide mechanistic insight how the transient desumoylation of IRF2BP is triggered and how SUMOylated IRF2BP limits expression of DUSP1. I appreciate that it is quite challenging to address these points, but I would suggest that prior to publication the authors do at least some candidate-based experiments to better define the SUMO-dependent repressive effect.

Major point:

1) As outlined above the manuscript lacks mechanistic data on the SUMO-dependent repression of DUSP1. SUMO-dependent repression of gene expression is a common concept and at least in some cases it has been demonstrated that it relies on SUMO-dependent promoter recruitment of co-repressors, including NCOR-SMRT-HDAC containing complexes. Since IRF2BP2 was found to be associated with NCOR, Sin3A and HDAC (see for example BioGRID database) it would be interesting to see whether any of these co-repressors is recruited to the DUSP1 promoter in a SUMO-dependent manner. This could be addressed by CHIP experiments in IRF2BP1 wild-type versus IRF2BP1 K579R expressing cells.

This is an excellent idea, and we started to test it. Unfortunately, although we could detect Sin3A, HDAC2 and NCOR1 on the DUSP1 and better on the ATF3 promoter by ChIP qPCR, promoter occupancy did not seem to alter significantly between wt and mutant cells or in regular HeLa cells that were starved and treated with or without EGF for 15 min. As these experiments had quite a number of technical problems, for example lack of synchrony of deSUMOylation upon EGF treatment and variable knockdown efficiency in the stable cell lines, we also tested the effect of E1 inhibition. But in this case, HDAC2 and Sin3A seemed to increase rather than decrease on the promoter. This may actually be consistent with our findings during the course of this revision, that the E1 inhibitor impairs - rather than stimulates transcription of DUSP1 (see above, Figure 1). We hope to follow up on this question in the future with DUSP1 and ATF3 reporter constructs that will allow more robust readouts.

2) The authors propose that IRF2BPs are rapidly deSUMOylated upon EGF stimulation. Could the involvement of distinct SUMO deconjugases in this process be tested by knock-down experiments? If feasible a distinct deconjugase might be identified and its differential interaction with IRF2BPs upon EGF treatment could be tested. This could strengthen the switch model, but would indeed need considerable experimental input.

We thank the reviewer for the suggestion. To begin to address this, we knocked down the SUMO isopeptidases of the Senp / Ulp family, Senp1, 2, 3, 5, 6 and 7 in HeLa cells stably overexpressing wildtype IRF2BP1. In these cells, knockdown of Senp3, but not of other isopeptidases, showed a clear effect. However, Senp3 depletion increased not only the SUMOylated form of IRF2BP1 but also the unmodified form, making interpretations rather difficult. We will obviously follow up on these observations, but this is clearly beyond the scope of this manuscript.

[Figures for referees not shown.]

Minor points:

1) According to the methods section the microarray experiment was done in triplicate for each cell-line, but Figure 3A does text [Figures for referees not shown.] not show any error bars. Is this Figure based on only one selected dataset? One would rather like to see the mean of all three experiments with error bars.

Here we used a representation that is commonly used in microarray analyses. We measured intensity values for each gene in the wt and mutant cell lines with and without EGF in triplicate. Each value was transformed to its log₂ value, then an average was formed for each triplicate. The log₂-fold-change was then calculated by subtracting the value obtained for the -EGF from the value obtained for +EGF in both cell lines. Hence there is no error bar.

But of course all samples were subjected to statistical analysis, to evaluate whether the fold changes observed between -EGF vs. +EGF are robust and reproducible. The parameter calculated for this is the FDR value. The result of this test can be found in the microarray table.

We have changed the figure legend in a way that it is hopefully more clear, and we added this information in the methods section.

Dear Frauke,

Thank you for the submission of your revised manuscript to EMBO reports. We have now received the reports from referee 1 and 2, who were asked to evaluate the revised version (copied below).

As you will see, both referees find that the study is significantly improved during revision and recommend publication.

Browsing through the manuscript myself, I noticed a few editorial things that we need before we can proceed with the official acceptance of your study.

- Your manuscript will be published in our "Reports" section. This requires that the Results and Discussion sections are combined. This will also help to shorten the manuscript text by eliminating some redundancy that is inevitable when discussing the same experiments twice to get closer to our character limit of 25,000 plus/minus 2,000 (main text, excluding materials and methods)

- Please reduce the number of keywords to 5. I suggest keeping EGFR, SUMO, IRF2BP1, DUSP1, ATF3.

- Reference format: please use et al when there are more than 10 authors, i.e., list the first 10 authors followed by et al

- Please add callouts to the panels of Figs. EV1 and EV4.

- Datasets: Please add the file name (Dataset EVx) into the .xls file. You could e.g., add this information into the "legend" tab.

- Table EV1: Please remove the legend from the manuscript file and add it to the table itself.

- Please change "Summary" to "Abstract" and describe all new findings in the Abstract in present tense.

- Please correct "Methods" to "Materials and Methods".

- I attach to this email a related manuscript file with comments by our data editors. Please address all comments and upload a revised file with tracked changes with your final manuscript submission.

- Finally, EMBO reports papers are accompanied online by A) a short (1-2 sentences) summary of the findings and their significance, B) 2-3 bullet points highlighting key results and C) a synopsis image that is 550x200-600 pixels large (width x height) in .png format. You can either show a model or key data in the synopsis image. Please note that the size is rather small and that text needs to be readable at the final size. Please send us this information along with the revised manuscript.

With kind regards,

Martina

Referee #1:

The authors have addressed my concerns.

Referee #2:

I'm totally satisfied with authors' replies to my previous concerns. I also thank the authors for their effort in these very difficult moments. In my opinion, the manuscript is suitable for publication.

The authors have addressed all minor editorial requests.

Prof. Frauke Melchior
University of Heidelberg
ZMBH
Im Neuenheimer Feld 282
Heidelberg 69120
Germany

Dear Frauke,

I am very pleased to accept your manuscript for publication in the next available issue of EMBO reports. Thank you for your contribution to our journal.

At the end of this email I include important information about how to proceed. Please ensure that you take the time to read the information and complete and return the necessary forms to allow us to publish your manuscript as quickly as possible.

As part of the EMBO publication's Transparent Editorial Process, EMBO reports publishes online a Review Process File to accompany accepted manuscripts. As you are aware, this File will be published in conjunction with your paper and will include the referee reports, your point-by-point response and all pertinent correspondence relating to the manuscript.

If you do NOT want this File to be published, please inform the editorial office within 2 days, if you have not done so already, otherwise the File will be published by default [contact: emboreports@embo.org]. If you do opt out, the Review Process File link will point to the following statement: "No Review Process File is available with this article, as the authors have chosen not to make the review process public in this case."

Please note that under the DEAL agreement of German scientific institutions with our publisher Wiley, your paper might be eligible for open access publication in a way that is free of charge for the authors. Please contact either the administration at your institution or our publishers at Wiley (emboreports@wiley.com) for further questions.

<https://authorservices.wiley.com/author-resources/Journal-Authors/open-access/affiliation-policies-payments/institutional-funder-payments.html>

Should you be planning a Press Release on your article, please get in contact with emboreports@wiley.com as early as possible, in order to coordinate publication and release dates.

Thank you again for your contribution to EMBO reports and congratulations on a successful publication. Please consider us again in the future for your most exciting work.

Kind regards,

Martina

Martina Rembold, PhD
Senior Editor

EMBO reports

THINGS TO DO NOW:

You will receive proofs by e-mail approximately 2-3 weeks after all relevant files have been sent to our Production Office; you should return your corrections within 2 days of receiving the proofs.

Please inform us if there is likely to be any difficulty in reaching you at the above address at that time. Failure to meet our deadlines may result in a delay of publication, or publication without your corrections.

All further communications concerning your paper should quote reference number EMBOR-2019-49651V3 and be addressed to emboreports@wiley.com.

Should you be planning a Press Release on your article, please get in contact with emboreports@wiley.com as early as possible, in order to coordinate publication and release dates.

YOU MUST COMPLETE ALL CELLS WITH A PINK BACKGROUND ↓
PLEASE NOTE THAT THIS CHECKLIST WILL BE PUBLISHED ALONGSIDE YOUR PAPER

Corresponding Author Name: Prof. Frauke Melchior

Journal Submitted to: EMBO

Manuscript Number: EMBOR-2019-49651-T